

# Assessment of Evolution of Mountain Lakes and Risks of Glacier Lake Outbursts in the Djungarskiy (Jetysu) Alatau, Central Asia, using Landsat Imagery and Glacier Bed Topography Modelling

Vassiliy Kapitsa[1], Maria Shahgedanova[2], Horst Machguth[3,4], Igor Severskiy[1], Akhmetkal Medeu[1]

[1] Institute of Geography, Almaty, Kazakhstan

[2] University of Reading, Reading, UK

[3] Department of Geography, University of Zurich, Zurich, Switzerland

[4] Department of Geosciences, University of Fribourg, Fribourg, Switzerland

Correspondence to: Maria Shahgedanova (m.shahgedanova@reading.ac.uk), Vassiliy Kapitsa
(vasil.geo@mail.ru)

**Abstract**

Changes in the abundance and area of mountain lakes in the Djungarskiy (Jetysu) Alatau between 2002

and 2014 were investigated using Landsat imagery. In 2002 and 2014, 599 lakes with a combined area of $16.26\pm0.85$ km$^2$ and 636 lakes with a combined area of $17.35\pm0.92$ km$^2$ respectively were identified. The number of lakes and their combined area increased by 6.2 % and 6.6 % representing growth rates of 0.51 % a$^{-1}$ and 0.55 % a$^{-1}$. Contact lakes exhibited the largest growth. Fifty lakes, whose potential outburst can damage existing infrastructure, were identified. The GlabTop2 model was applied to simulate ice

thickness and subglacial topography using glacier outlines for 2000 and SRTM DEM as input data achieving realistic patterns of ice thickness. 513 overdeepening in the modelled glacier beds, presenting potential sites for the development of lakes, were identified with a combined area of 14.7 km$^2$. Morphometric parameters of the modelled overdeepenings were close to those of the existing lakes. A comparison of locations of modelled overdeepenings and newly formed lakes in the areas de-glacierized

in 2000-2014 showed that 67 % of the lakes developed at the sites of the modelled overdeepenings. The rates of increase in areas of new lakes correlated with areas of modelled overdeepenings. Locations where





hazardous lakes may develop in the future were identified. The GlabTop2 approach is shown to be a useful tool in hazard management providing data on the potential evolution of future lakes.

## 1. Introduction

The retreat of mountain glaciers has been observed since the end of the Little Ice Age (LIA) worldwide and has intensified during the last 30-50 years. The mountains of Asia, including the Pamir, Tien Shan and Djungarskiy (Jetysu) Alatau, where currently glaciers occupy about 16,427 km$^2$ (Sorg et al., 2012), are no exception. Across this region, glaciers retreated losing both area and mass (Solomina et al., 2004;

Shahgedanova et al., 2010; Kutuzov and Shahgedanova, 2009; Sorg et al., 2012; Yao et al., 2012; Farinotti et al., 2015; Pieczonka and Bolch, 2015) with the highest retreat rates reaching 1 % a$^{-1}$ reported for the northern Tien Shan and Djungarskiy Alatau (Narama et al., 2010 a; Severskiy et al., 2016).

One of the main impacts of glacier recession is an increase in number, area and volume of glacial and proglacial lakes. These lakes are potentially dangerous as rapid snow and glacier melt can result in lake

outbursts threatening life and infrastructure (Richardson and Reynolds, 2000). Many studies of glacial and proglacial lake evolution were published for Asia, focussing on the following three topics: (i) lake inventories documenting spatial and temporal trends in lake formation and development; (ii) identification of potentially hazardous lakes and assessment of likelihood of glacier lake outburst (GLOF); and more recently (iii) attempts to predict the formation of lakes following de-glaciation.

To date, these studies were mostly concerned with the Hindu-Kush – Himalaya region and the Tibetan Plateau (e.g. Bolch et al., 2008; Komori, 2008; Ye et al., 2009; Huang et al., 2011; Gardelle et al., 2011; Wang et al., 2012; Zhang et al., 2011; 2013; Fujita et al., 2013). Most reported an increase in number of lakes, their areas and water storage (e.g. Wang et al., 2012). Many studies highlight important regional differences in lake evolution (e.g. Gardelle et al., 2011; Zhang et al., 2013) showing that trends in changes

in lake areas and volumes correlate well with changes in glacier mass balance and other factors such as changes in evaporation, water withdrawal for human use in the water-deficient regions and lake drainage



(either deliberate or natural). Possible future development of lakes in the Karakorum – Himalaya was investigated by Linsbauer et al. (2012; 2016) who applied the Glacier Bed Topography (GlabTop) model to identify overdeepenings in the glacier beds in which potential lakes may evolve in the future.

Glacial and proglacial lakes in the mountains of Central Asia received less attention. Recently, an inventory of lakes with individual areas in excess of 2000 $m^2$ was compiled by Wang et al. (2013) for the Tien Shan reporting an increase in the count and combined area of lakes of 22.5 % and 16.7±2.9 % respectively between 1990 and 2010. Wang et al. (2015) assessed water storage in lakes of the Tarim basin reporting a decline in the levels of glacier-fed lakes despite the observed glacier thinning and

negative mass balance and attributed this discrepancy to water abstraction for agricultural needs. Bolch et al. (2011) assessed hazard potential of 132 lakes in the Zailiiskiy Alatau. Other studies, including Narama et al. (2010b),  Jansky et al. (2010), Medeu et al. (2013) and Herget et al. (2013), and earlier studies from the 1970s-1990s reviewed by Medeuov et al. (1993) and more recently by Evans and Delaney (2015) focused on hazard potential of individual lakes particularly in the densely populated Zailiiskiy Alatau.

The Djungarskiy Alatau is one of the mountain regions of Central Asia where lakes are particularly widespread and where glacier retreat rates are among the highest in the region (Severskiy et al., 2016). Between 1955 and 2011, 48 % of the glacier area was lost; between 2000 and 2011, glaciers lost 12 % of their combined area (Table  S1) retreating at a rate of 1.1 % $a^{-1}$. Vilesov et al. (2013) reported a widespread degradation of permafrost and melt of buried ice across the region. Lake dams are predominantly

composed of morainic material and thawing of permafrost and buried ice increase the risk of moraine failure and lake outburst (Popov, 1986; Jansky et al., 2010; Bolch et al., 2011). A number of GLOF events, followed by mudflows were reported in the region in the 1970s and early 1980s, when positive temperature anomalies were close to those observed in the 2010s, e.g. in the upper reaches of the Aksu River in 1970 and 1978 and the Sarkand valley in 1982 (Medeuov et al., 1993). In 1982, the outburst of

Lake Akkol in the upper reaches of the Sarkand River at about 3200 m a.s.l. and its discharge into Lake Tranzitnoe and then Lake Kokkol, which were overtopped, led to the formation of the largest recorded





mudflow in the region with an estimated volume of water and transported material of approximately 2.7 million $m^3$. The maximum discharge was estimated as 2300 $m^3\,s^{-1}$. Within the town of Sarkand, located 45 km downstream, discharge reached 300 $m^3\,s^{-1}$ leading to a widespread destruction of infrastructure

(Tikhomirov and Shevyrtalov, 1985).

Yet despite the rapid retreat of glaciers, abundance of lakes and recently initiated development projects and expanding infrastructure, there is no comprehensive inventory of lakes in the region enabling assessment of their evolution and hazard potential. An inventory for a single year of 2002 was compiled using Landsat imagery by Severskiy et al. (2013) for selected regions in the Djungarskiy Alatau reporting

that the majority of lakes were located between 3300 m a.s.l. and 3600 m a.s.l. on either young (20[th] or 21[st] century) or LIA and older moraines. Medeu et al. (2013) examined changes in 48 lakes in the catchment of the River Khorgos between 1978 and 2011 using the USSR 1:100,000 topographic maps and Landsat imagery. This study concluded that number and combined area of lakes positioned on the 20-21[st] century moraines in proximity to glaciers exhibited the largest change while those positioned on the

older moraines changed least. The study stressed that opposite trends in lake evolution can be observed within the same relatively small region particularly with regard to lakes positioned on the young moraines. Kokarev and Shesterova (2014) assessed changes in glacier area in the southern Djungarskiy Alatau where they identified 190 lakes with the total area of 6 $km^2$ as in 2000 but did not provide any analysis of lake distribution and evolution.

The aim of this paper is to provide a comprehensive inventory of lakes and their observed and future evolution in the Djungarskiy Alatau and assessment of their hazard potential using remote sensing and GlabTop version 2 model. The specific objectives are to (i) present inventories of lakes for 2002 and 2014; (ii) assess changes in number and area of lakes between these years for different river basins and elevation, types and size classes; (iii) assess hazard potential of lakes providing recommendations for further

investigation using higher-resolution satellite imagery and field studies; (iv) model glacier-bed



topography and identify overdeepened areas where future lakes may develop testing the results against the observed development of lakes.

## 2. Study Area

The Djungarskiy Alatau is located at the north-eastern flank of the Tien Shan (Fig. 1). The maximum elevation is 4622 m but most peaks extend to about 3800 m. The glacier snouts descend to approximately 3400 m a.s.l.  In 1956, the combined glacierized area was 814 km$^2$ declining to 465 km$^2$ in 2011 (Vilesov et al., 2013). By 2011, 103 glaciers (a half of those catalogued in 1956) had disappeared. At present, the northern sector (basins of the Bien, Aksu and Lepsy rivers) is most heavily glacierized followed by the

southern (basins of the Horgos and Usek) and then western (the Karatal basin) sectors. The number and combined area of glaciers decline towards the east (basins of the Tentek, Tastau and Rgaity) (Fig. 1; Table S1).

The climate is characterised by strong seasonal contrasts in temperature and precipitation (Fig. 2). In winter, the region is dominated by the western extension of the Siberian anticyclone which predetermines

low temperatures and small amounts of precipitation. The westerly flow dominates in autumn and spring with frequent depressions and precipitation maxima in October-November and April-May. In summer, the thermal Asiatic depression dominates and the advection of warm, dry air from the south results in low precipitation. Precipitation declines towards the east from 1400-1600 mm at the altitude of 3400-3600 m a.s.l. in the west to about 1000 mm in the east (Vilesov et al., 2013). The accumulation period extends

between mid-September and early June; ablation period is limited to June-July-August (JJA).

## 3. Data and Methods

### 3.1. Satellite and ground-based data





To map glacier lakes, six [nearly] cloud-free Landsat scenes for 2002 and 2014 were obtained from the

US Geological Survey (USGS; http://glovis.usgs.gov/) in the Universal Transverse Mercator (UTM) zone

44 WGS 84 projection (Table 1). Five out of six scenes, were obtained in August to minimize uncertainty

associated with seasonal fluctuations of lake levels. According to the data from the meteorological station

Lepsy located at 1012 m a.s.l. (Fig. 1), the JJA temperatures in 2002 and 2014 were very close at 16.9ºC

and 17.3ºC respectively insuring similar melt conditions. The selection of the July-August scenes helped

130 to eliminate one of the main sources of uncertainty in lake mapping – the presence of seasonally frozen

water and seasonal snow on the frozen surfaces of lakes. It also ensured that lake levels were at their

maximum aiding mapping of the contact lakes and lakes located on the young moraines whose areas may

be significantly reduced during the cold season. A collection of the ground-based and oblique aerial

photographs (e.g. Fig. 3, 4) obtained in 2014 was used as supplementary data. ASTER GDEM2

135 (https://asterweb.jpl.nasa.gov/gdem.asp) was used to derive data on elevation and slope angles.

### 3.2. Lake identification

599 and 636 lakes were mapped on the 2002 and 2014 images respectively using ERDAS Imagine. All

lakes had natural regime and were not lowered artificially or used for water abstraction. A number of

previous studies developed (Huggel et al., 2002; Li and Sheng, 2012) and applied (Ye et al., 2009; Bolch

et al., 2011; Gardelle et al., 2011; Wang et al., 2012; 2013) automated technique for the mapping of lakes

using the Normalised Difference Water Index (NDWI) based on the low water reflectance in the NIR

band and using various band combinations (e.g. green, blue). The well-known advantages of automated

mapping are (i) the reproducibility of results and (ii) the ability of the method to quickly map large

numbers of lakes. The disadvantages are (i) inability to map small lakes imposed by resolution of

multispectral Landsat imagery; (ii) misclassification of lakes due to shadows (although this problem can

be negated by the use of a shadow mask generated using a precise DEM (Huggel et al., 2002)) and (iii) a

wide range of NDWI values characterising lake pixels resulting from the widely different physical and



chemical properties of mountain lakes and, consequently, a wide range of their spectral signatures making

application of a single threshold to mapping of mountain lakes problematic (Li and Sheng, 2012). Lakes

with high water turbidity are especially difficult to map using automated techniques (Bolch et al., 2011)

as well as small lakes which tend to have low NDWI values (Li and Sheng, 2012).

In this study, automated mapping of lakes was initially performed using various band combinations but

did not produce satisfactory results because the method frequently misclassified melting glaciers as lakes

and importantly, consistently failed to distinguish lakes with high water turbidity which is particularly

typical of the lakes developing on newly formed moraines (Type 2 lakes as defined in Sect. 3.4; Fig. 3).

In 2014, there were 234 Type 2 lakes in the study area (37 % of all lakes) and the high proportion of

misclassified lakes of this type alone significantly reduced the advantages of the automated mapping with

regard to efficiency and reproducibility of the results.

Moreover, small lakes cannot be mapped using automated techniques. Applying NDWI to the Landsat

imagery with 30 m resolution, Wang et al. (2013) used a threshold of 2000 $m^2$ in digitisation of lake areas

as it covered approximately three pixels of uninterrupted water body. In the Djungarskiy Alatau, 7 % of

all lakes had individual areas less than 2000 $m^2$ (Sect. 4, 5). The identification and mapping of smaller

lakes was important because, in the study area, they frequently form vertical sequences or cascades

whereby several lakes either have hydraulic connection or are located in close proximity at different

elevations (e.g. Fig. 3). While an outburst of a single small lake is unlikely to cause severe damage, it can

trigger an outburst of a larger or multiple lakes. Having reviewed a number of studies documenting debris

flows in the northern Tien Shan, Evans and Delaney (2015) commented that outbursts of small lakes can

initiate debris flows whose volumes significantly exceed those of the initial outburst. In addition, the

detection of small lakes may be used in the context of interpretation of future lake development alongside

the outputs from GlabTop2 model. Therefore, lakes were mapped manually using channels 7, 4, 2 (Li and

Sheng, 2012) and panchromatic channel 8. The use of the panchromatic channel with 15 m resolution,

which requires manual mapping, enabled us to lower the threshold of digitisation from 2000 $m^2$ to 700



m$^2$. The outcomes of the attempted automated classifications, where successful, were used as axillary

material.

### 3.3. Quantification of uncertainty

We considered uncertainties of lake area mapping in individual years and uncertainties of lake area change

between 2002 and 2014. The main uncertainty in lake area identification in a single year is the uncertainty

originating from the lake boundary delineation by an individual operator. To quantify it, we followed the

multiple digitization method proposed by Paul et al. (2013) for the digitization of glaciers. Seventy one

lakes with areas ranging between 700 m$^2$ and 190,000 m$^2$ were mapped by four operators using the Landsat

scenes from 2002 and 2014. For each lake in a given year, the mean and standard deviation of four

measurements were calculated and the uncertainty value was calculated as a ratio between standard

deviation and mean area multiplied by 100 %. These were averaged over three size classes. The following

uncertainty values were applied to the 2002 and 2014 measurements respectively: ±8.2 % and ±8.7 % for

lakes with individual areas less than 10,000 m$^2$; ±6.9 % and ±6.7 % for lakes with areas between 10,000

m$^2$ and 50,000 m$^2$; and ±4.3 % and ±4.0 % for lakes larger than 50,000 m$^2$. The mean value of uncertainty

was ±6.9 %.

To quantify the uncertainty of measurements of lake area change, uncertainties due to image co-

registration and lake boundary delineation were considered. The scenes from 2002 and 2014 were co-

registered and for each pair of scenes, a network of 10-15 tie points was established using clearly

identifiable terrain features whose location did not change. The derived root-mean-square error (RMSE$_{x,y}$)

values ranged between ±3.7 m and ±5.3 m. The uncertainty term was calculated following a method

proposed by Granshaw and Fountain (2006) and modified by Bolch et al. (2010). On the individual scenes,

a buffer with a width of half of RMSE$_{x,y}$ was created along the lake boundaries and the uncertainty term

was calculated as an average ratio of the original lake areas to the areas with a buffer increment. The total

uncertainty value was calculated as root mean square of the uncertainty values of co-registration for 2002

and 2014 and lake boundary identification for 2002 and 2014. The mean value of the total uncertainty of

lake area change was ±15.7 %.

## 3.4. Classification of lakes

Various classifications of lakes exist in literature including those developed for the northern Tien Shan.

A classification developed by Popov (1986) comprised four types and ten sub-types of lakes including

supra-glacial, proglacial contact, glacial-morainic and morainic lakes with the two latter located on newly

formed and LIA or older moraines respectively. A classification by Medeuov et al. (1993) included

proglacial contact, thermokarst, moraine-dammed and cirque lakes (the latter forming in the ice-free

glacier cirques). This classification was later modified by Medeu et al. (2013) to distinguish between lakes

forming on the 20-21$^{st}$ century moraines and on the LIA or older moraines and to include rock-dammed

lakes.

In this study, all mapped lakes were assigned to one of following types (Fig. 4): Type 1 - contact lakes

developing at glacier tongues; Type 2 - proglacial morainic lakes forming on the 20-21$^{st}$ century moraines

in close proximity (typically within 500 m) to but without contact with glacier tongues; Type 3 - morainic

lakes positioned in depressions on the LIA or older moraines; Type 4 - dammed lakes forming due to the

damming of rivers and streams by rocks. There are no ice-dammed lakes in the region. This classification

is less detailed than those by Popov (1986) and Medeuov et al. (1993), however, analysis of Landsat

imagery does not enable a more detailed discrimination. Medeu et al. (2013) showed that including two

types of morainic lakes is important because their responses to climate and glacier change are different.

Type 2 lakes often have hydraulic connections to glaciers and moraines, on which they develop, usually

contain buried ice (Vilesov et al., 2013). Both factors control rapid response of these lakes to temperature

increase. Older moraines may be underlain by permafrost whose response to temperature change is slower



(Severskiy, 2009) but whose melt can potentially lead to dam instability (Bolch et al., 2011). The dammed

lakes are currently not in contact with glaciers and direct discharge of melt water into these lakes is

unlikely but may occur via discharge from the Type 1-2 lakes located upstream potentially initiating

outflow from a dammed lake. An example is Lake Kazankol in the Khorgos river basin (Fig. 4d) with a

volume of 6 million m$^3$ upstream of which four lakes are located (Medeuov et al., 1993).

**3.4. Assessment of risks of lake outburst**

Several studies proposed criteria for the identification of potentially dangerous lakes (e.g. Huggel et al.

2002; 2004; Allen et al., 2009; Bolch et al., 2008; 2011). We followed the three-tier methodology of

assessment of hazard potential of lakes proposed by Huggel et al. (2002) and, having completed their

Level 1 (basic detection of lakes), we focused on Level 2:  Consideration of criteria which can be derived

from Landsat imagery and ASTER GDEM2. These criteria and the order of their consideration were: (i)

lake type; (ii) presence of cascade of lakes; (iii) lake area (as an indicator of peak discharge); (iv) distance

to infrastructure; (v) average slope; (vi) presence of slopes of 45$^o$ and steeper in proximity to the lake.

Single lakes and cascades of lakes (hydrologically connected or located in close proximity) were assessed

separately with regards to criteria (iii) and (iv). We did not include change in areas of individual lakes as

a criterion of hazard. While increase in lake area is a factor making lakes outburst more likely (Bolch et

al., 2011), no change or reduction in lake area is not a guarantee that outburst will not occur because

potential thawing of ice contained within the morainic dam (Jansky et al., 2010; Herget et al., 2013; Evans

and Delaney, 2015) or blockage of channels within the dam (Narama et al., 2010b) can lead to its breach

in a short period of time.

We considered as potentially hazardous contact lakes (Type 1) and those located on the young moraines

(Type 2) on the basis of the analysis of lake evolution according to their type presented in Section 4. Using

data on lake areas from 2014, we disregarded all single lakes whose discharge is unlikely to generate large



flood events. Lake area is frequently used as a proxy for peak discharge (Huggel et al., 2002), however, there is no uniformly accepted threshold for a critical lake area and these are often set according to the previous GLOF events (e.g. Cook et al., 2016). Wang et al. (2013) used a threshold of 100,000 m$^2$ assessing potentially dangerous lakes across the Tien Shan while the Kazakhstan State Agency for

Mudflow Protection (KSAMP) uses a threshold of 10,000 m$^2$ in the Zailiiskiy Alatau (Bizhanov et al., 1998) but does not specify any threshold for the study region. For single lakes, we set a threshold of 20,000 m$^2$ based on the consideration that the lakes in the Aksu and Kora valleys whose previous outbursts significantly damaged the infrastructure had areas of approximately 20,000 m$^2$ (Medeuov et al., 1993) and that the valleys in the Djungarskiy Alatau are approximately twice the lengths of the valleys in the

Zailiiskiy Alatau. We considered all Type 1 and Type 2 lakes, irrespective of their size, as potentially dangerous if they were part of a cascade of lakes with further lakes located on the potential flood path.

Lakes were considered as potentially dangerous if they had hydrological connection to the downstream settlements, infrastructure and agricultural fields located within 60 km distance. In the absence of flow modelling, the selection of this threshold is based on the past events when lake outburst in the Aksu and

Sarkand valleys generated flows travelling up to 45 km from their sources respectively and the fact that areas of these lakes increased in 2014 in comparison to the 1970s-1980s when the outbursts occurred. In case of lake cascades, the distance was measured from a lake positioned at the lowest elevation in the cascade. The locations of infrastructure objects and agricultural fields was derived from the 2014 Landsat imagery. Pastures were not included as they could not be distinguished from any other natural grasslands.

Haeberli (1983) and Huggel et al. (2002) suggested an average slope threshold of 11º as a condition for debris flow propagation. In the events of previously recorded mud- and debris flows in the Djungarskiy Alatau, which travelled distance over 40 km, the average incline was 6-9º (Tikhomirov and Shevyrtalov, 1985; Medeuov et al., 1993) but very few events were studied. Allen et al. (2009), Frey et al. (2010) and Bolch et al. (2011) used average incline of 3º as a threshold.  Given the uncertainty in determination of



slope threshold resulting from multiple approximations based on limited data, we assumed the most

stringent threshold of 3º.

Mass movements and in particular ice falls and calving events can serve as triggering mechanisms for

lake outburst (Evans and Delaney, 2015) and it has been suggested that a 45º threshold represents a critical

value for slopes (Alean, 1985; Bolch et al., 2011; Cook et al., 2016). A slope map was generated from

ASTER GDEM2 to identify slopes of 45º and steeper. Cook et al. (2016) reviewed typical runout distances

for ice avalanches and rock falls concluding that most widely used values vary between $10^2$ and $10^3$ m and

used 500 m as a threshold for runout distance. On the basis of this review, we considered lakes located

within 500 m of slopes exceeding 45º threshold as dangerous. Snow avalanches were not considered as

capable of causing lake outburst as these occur during the time when both lakes and the ground are frozen

preventing the development of floods capable of travelling long distances even if the lake ice is broken

by snow mass (Popov, 1986).

In order to assess the severity of potential GLOF events, lake volumes were estimated using an empirical

relationship between lake area and volume derived from bathymetric measurements of 32 Type 1 and

Type 2 lakes, with areas ranging between 2000 $m^2$ and 200,000 $m^2$, in the Zailiiskiy Alatau and

Djungarskiy Alatau in 2009-2014 at the end of the ablation seasons in late August-September (Medeu and

Blagoveshenskiy, 2015). In these surveys, lake depth was measured using echo sounding conducted along

the multiple profiles spaced at approximately 10 m with the along-profile increment of 3-5 m. The values

of lake volumes were derived from the depth measurements using Surfer software. The high-density depth

measurements of each lake enabled representation of complex lake morphometries which are often

neglected in lake area-volume scaling (Cook and Quincey, 2015).  The relationship between volumes and

areas of the surveyed lakes is shown in Fig. 5 and is approximated by the Eq.(1):

V = 0.036 * A$^{1.49}$                                                                                      (1)



where V (m³) is lake volume and A (m²) is lake area. This relationship is almost identical to that quoted

by Evans (1986) for the ice dammed lakes in Canada. Peak discharge was estimated using the relationship

between lake volume and discharge of Haeberli (1983):

$$Q_{max} = 2V / t \qquad\qquad (2)$$

where V is volume calculated using Eq. (1) and t is time equalling 1000 s. Peak discharge was calculated

for the individual lakes which were identified as potentially hazardous. While peak discharge represents

the worst-case scenario expected from a complete drainage of individual lakes, which is unlikely in case

of larger lakes, it does not account for potential drainage of cascades of multiple lakes as observed in the

region in the 1970s-1980s.

### 3.5. Modelling glacier bed topography and detection of overdeepenings

Here we use the Glacier Bed Topography (GlabTop2) model to derive glacier bed topography based on

glacier surface slope. GlabTop2 is fully described in Frey et al. (2014) and is based on the same concept

as the original GlabTop model (Linsbauer et al., 2012). Both models assuming a constant basal shear

stress along the central flowline of an entire glacier. Ice thickness is then calculated according to

$$h = \frac{\tau}{f \cdot \rho \cdot g \cdot \sin\alpha} \qquad\qquad (3)$$

where h is ice thickness (m), $\tau$ is basal shear stress, f is shape factor (0.8), $\rho$ is ice density (900 kg m⁻³), g

is acceleration due to gravity (9.81 m s⁻²) and $\alpha$ is glacier surface slope. The basal shear stress is estimated

from an empirical relationship between $\tau$ and glacier height extent $\Delta z$ (i.e. maximum elevation minus

elevation of the glacier tongue) according to Haeberli and Hoelzle (1995). Here we calculated $\Delta z$ using

the elevation $z$ of the lowest and highest grid cell that fall within each glacier polygon.

GlabTop2 is fully automated, entirely grid-based and first calculates the ice thickness at a set of randomly

selected grid areas. Subsequently, these thickness values are interpolated to the entire glacier area. To





achieve realistic glacier cross-sections, the interpolation scheme assigns a minimum, non-zero ice thickness to all grid cells directly adjacent to the glacier margin. The method includes some non-physical, tunable parameters such as the density of the random point sampling (see Frey et al. (2014) for a detailed description and discussion).

To calculate glacier bed topography of all glaciers in the Djungarskiy Alatau, identical settings as in Frey et al. (2014) were used. Ice thickness distribution for all glaciers larger 0.1 km$^2$ has been calculated based on the outlines of glaciers as in 2000 generated as part of regular cataloguing of glaciers of Kazakhstan (Severskiy et al., 2016) and SRTM DEM surface topography (https://lta.cr.usgs.gov/SRTM1Arc). To reduce the influence of small-scale surface undulations in the ice thickness modelling, SRTM DEM with

30 m resolution was down-sampled to 75 m cell size.

Overdeepenings in the glacier beds, where future lakes may develop, were identified following the methodology outlined in Linsbauer et al. (2012; 2016) namely a sink fill algorithm available in ARC GIS 10.4 was run on the modelled bed topography to create a sink filled version of DEM and the original bed topography DEM was subtracted from the sink filled version. The difference grid between the filled DEM

and the initial DEM without glaciers resulted in a bathymetry raster of the overdeepenings. The following morphometric characteristics of overdeepenings were derived: area, maximum and mean depth, volume, maximum length, maximum width (perpendicular to the longest line), and elongation (width-to-length ratio) following Linsbauer et al. (2016) and Haeberli et al. (2016). Overdeepenings with individual areas larger than 11,000 m$^2$, corresponding to the area of approximately two grid cells in the glacier bed

topography, were considered to exclude potential model artefacts in line with previous studies (Linsbauer et al., 2012; Haeberli et al., 2016).

## 4. Results

### 4.1. Characteristics of lakes and their distribution




In 2002 and 2014, 599 lakes with a combined area of 16.26±0.85 km$^2$ and 636 lakes with a combined area

of 17.35±0.92 km$^2$ respectively were identified. In 2002, the largest measured lake area was 1.03 km$^2$, the

smallest 0.0007 km$^2$ and the mean was 0.027 km$^2$. Lakes with individual areas of 0.002-0.05 km$^2$

prevailed accounting for 80 % by number and 42-45 % of the combined area (Fig. 6 a, b).

The lakes were positioned within the elevation bands of 2220-3660 m and 2220-3690 m in 2002 and 2014

respectively. The small shift of the upper boundary of 30 m is close to the absolute vertical accuracy of

ASTER GDEM2 of 17 m (Meyer et al., 2011). The majority of lakes (60 % and 56 %) accounting for 56

% and 52 % of their total combined area in 2002 and 2014 respectively, were positioned between 3100 m

and 3400 m a.s.l. (Fig. 7).

The Type 3 lakes prevailed by both number and combined area followed by the Type 2 lakes (Table 2).

There were relatively few Type 4 lakes but their individual areas were an order of magnitude larger

averaging 0.11 km$^2$ than the other types of lakes. Type 4 lakes were positioned at lower elevations and

for this reason, the average lake size within the lowest elevation range of 2200-2500 m a.s.l. was 0.2 km$^2$

which is higher than in any other elevation band.

The largest number of lakes with the highest total combined area occurred in the catchments of the rivers

Usek, Koksu and Aksu of which the first two are the largest catchments in the study area (Table S2; Fig.

1). The average size of the lakes in these catchments was 0.023-0.036 km$^2$. Larger lakes with mean areas

of 0.04 km$^2$ and 0.062 km$^2$ were characteristic the Khorgos and Lepsy catchments respectively. The

number and combined area of lakes in 15 catchments as in 2002 (Table S2) correlated closely with the

number of glaciers (correlation coefficients of 0.86 for both) and with combined glacierized area (0.67

and 0.72 respectively) (Table S1).

**4.2. Changes in the number and characteristics of the lakes**



Between 2002 and 2014, the overall number of lakes and their combined area increased by 6.2 % and 6.6 % respectively. While areas of 19.5 % of the lakes increased; areas of 12.5 % of the lakes declined including those lakes that have drained completely (Table 3; Fig. 6 c, d). Changes in 68 % of all the lakes (as in 2002) were within the uncertainty of measurement.

Fig. 6 illustrates changes in areas and abundance of lakes according to size category. The count of lakes increased in in all size categories except the largest ($>0.1$ km$^2$) where the count did not change and the smallest ($<0.002$ km$^2$) in which the total number of lakes declined because their areas increased in 2014 in comparison with 2002 and they were assigned to the next size category (Fig. 6a). A small increase in the combined area was observed in all categories too but was close to the uncertainty of measurements (Fig. 6b) and lakes, whose areas either increased (Fig. 6c) or decreased (Fig. 6d) beyond the uncertainty, were analysed separately. The smaller lakes exhibited greater expansion with the average area of the smallest lakes increasing by 220 % (Fig. 6c). The average area reduction was lower than increase in all size categories and did not appear to correlate with lake size (Fig. 6d).

The largest increase in abundance and average area, both absolute and relative, characterised Type 1 lakes (Table 4). Out of 44 lakes in the whole sample, which doubled their area, 31 belonged to this type. Out of 69 newly formed lakes, 51 were Type 1 lakes while very few Type 1 lakes decreased in size or drained completely (Table 4). The second largest category exhibiting growth was Type 2 lakes. Although average increase was smaller than that of the Type 1 lakes, their overall numbers and combined area increased by 30 % (Table 2) due to the transition of 35 lakes from the contact (Type 1) lakes (Table 4). The Type 2 lakes exhibited the most dynamic behaviour as, alongside their growth and formation of new lakes, 12 % and 11 % disappeared or decreased respectively between 2002 and 2014. The count and area of Type 3 and Type 4 lakes exhibited smaller variations.

Figure 8 shows locations of those lakes whose areas exhibited changes exceeding the uncertainty of measurement. The largest number of new lakes and those that increased in size were positioned in the northern and western sectors of the Djungarskiy Alatau in the basins of the rivers Aksu (10 % increase in





combined area), Bien (14 %), Kora (17 %) and Koksu (11 %) between approximately 3000 m and 3500

m a.s.l. The Koksu catchment accommodates the largest combined glacierized area in the region and the

Kora catchment is characterised by the highest proportion of glacierized area (Table S1). Out of 69 new

lakes, 25 formed in these four catchments (Fig. 8c; Table S2). The distribution of lakes whose areas

declined was uniform across region and most of these lakes were positioned at slightly lower elevations

of 3000-3300 m a.s.l.

**4. 3. Potentially dangerous lakes**

Fifty lakes were identified as potentially dangerous and most are located in the north-western and south-

western sectors of the study area (Fig. 1; Table S3). Three largest lakes had individual areas of about

200,000 m$^2$. Area of the second largest lake in the sample (N 5 in the Usek basin) increased by 29 %

between 2002 and 2014. The potential worst-case scenario peak discharge of the three largest lakes was

estimated to exceed 5,600 m$^3$ s$^{-1}$. Peak discharge of 35 lakes may exceed 300 m$^3$ s$^{-1}$ and this was the

registered velocity of the flow that devastated the town of Sarkand in 1982 following the outburst of Lake

Akkol (N 43) positioned 45 km upstream (Tikhomirov and Shevyrtalov, 1985).

The characteristic arrangement of lakes in vertical sequences or cascades creates potential for outburst of

multiple lakes. Therefore, even small lakes (e.g. 6 lakes with areas below 20,000 m$^2$; Table S3), which

are frequently disregarded in GLOF assessments, may be hazardous. Thirty two of the potentially

dangerous lakes are part of cascades of lakes including two largest lakes in the sample and those in the

Aksu (N 45) and Sarkand (N 43) valleys which outburst in 1970, 1978 and 1982 respectively causing

overtopping of the lakes located at lower elevations. Lake Kapkan (N 25), identified as dangerous and the

only lake in the Djungarskiy Alatau that has been periodically lowered since August 2014, belongs to a

cascade of six lakes. Four of those lakes (not identified as potentially dangerous by themselves), including

Lake Boskol (Type 3) with an estimated volume of about 5.34x10$^5$ m$^3$ and Lake Kazaknkol  (Type 4; Fig.



4d) with an estimated volume of $6 \times 10^6$ m$^3$, are located at lower elevations on the potential flow path to the town of Khorgos hosting the recently established International Trade Centre (Medeu et al., 2013).

Lakes N 15 (Type 1) and 17 (Type 2; 72 % increase) in the Aksu catchment are positioned at the top of

the largest in the Djungarskiy Alatau cascade of lakes (Fig. 3 and 9a). The largest lake (N 16) in this cascade has an area of 185,000 m$^2$. The outburst of Lakes 15 ad 17 can potentially cause overtopping of Lake 13 whose area increased by 121 % between 2002 and 2014 reaching 67,000 m$^2$. The outburst of Lake 13 and Lake 9, whose area increased between 2002 and 2014 by 57 % reaching 68,825 m$^2$, can trigger, in the worst-case scenario, an outburst of four downstream lakes with combined area exceeding

340,000 m$^2$.

A number of large cascades consisting of multiple lakes are positioned in the basin of the river Bien. Figure 10 shows a cascade comprised by four lakes of Types 1 and 2 with a combined area of 119,300 m$^2$. A contact Lake N 22, positioned at the top of the cascade, formed after 2002 reaching 44,200 m$^2$ in 2014, and has potential for further expansion (Sect. 4.4; Fig. 10b). In the Bien basin, the downstream

infrastructure and farmland are located within 23-30 km from the potentially dangerous lakes, which is closer than in other regions. However, this is the only region in the study area where overall slopes are small at 3-5° which may reduce the runout distance of floods and mudflows.

## 4.4. Ice thickness modelling and detection of overdeepenings

### 4.4.1. Ice thickness calculation

Ice thickness distribution for all glaciers larger 0.1 km$^2$ has been calculated based on the glacier outlines for 2000 and the SRTM DEM surface topography with original 30 m spatial resolution resampled to 75 m resolution. The total calculated ice volume in the Djungarskiy Alatau amounts to 29 km$^3$. Average modelled ice thickness is relatively shallow at 33 m; maximum ice thickness reaches 197 m. Both values

reflect the fact that the glaciers of the Djungarskiy Alatau are mostly small mountain-valley or cirque

glaciers. To our knowledge, no ice thickness measurements have been carried out in the Djungarskiy Alatau besides the extensive helicopter-borne measurements on 103 glaciers in 1981, published by Macheret et al. (1988). However, the hand-drawn maps and sketches of ice thickness distribution, shown in the aforementioned study, do only allow for a qualitative comparison to our model results. We find that the measurements and our model results agree on relatively thin glaciers with maximum ice thicknesses of typically around 80 to 120 m and rarely exceeding 150 m. The qualitative comparison agrees with the more quantitative evaluations where GlabTop2 has been shown to be well suited for mountain glaciers (Frey et al., 2014, Farinotti et al., In Press).

An extensive model intercomparison exercise (Farinotti et al., In Press) has compared ice thickness measurements from various glaciers to a variety of different modelling approaches that all calculate ice thickness based on glacier surface properties. The study has shown that for mountain glaciers both GlabTop and GlabTop2 perform similar to other approaches. The intercomparison exercise also showed that an uncertainty of ±30 % in modelled ice thickness, earlier established for GlabTop (Linsbauer et al., 2012) and GlabTop2 (Frey et al., 2014), is fairly typical for all modelling approaches. We thus suggest adopting the same uncertainty for the present modelling of the ice thickness distribution in the Djungarskiy Alatau.

In the context of this study, it would be of particular interest to compare modelled and measured overdeepenings underneath glacier tongues. However, Macheret et al. (1988) indicate lacking bed returns on the termini of the vast majority of the measured glaciers, making such a comparison virtually impossible. Instead, we analyse whether some of the overdeepenings modelled underneath glacier termini in 2000 have developed into actual lakes by the year 2014.

### 4.4.2. Detection of overdeepenings and potential development of lakes


513 overdeepenings with individual areas in excess of 11,000 m$^2$ were detected in the modelled glacier

beds within the glacierized area as in 2000. Their combined area was estimated as 14.7 km$^2$ which

corresponds to 3 % of the total glacierized area (Table S1).

Most overdeepenings are small and shallow with length and width of a few hundred meters and maximum

depth of 65 m (Fig. 11). The individual areas of 96 % of all overdeepenings are less than 0.1 km$^2$. Larger

overdeepening with individual areas between 0.05 km$^2$ and 0.5 km$^2$ are found in the regions where existing

lakes are most abundant (the Aksu, Usek and Khorgos basins) but also in the north-east of the region

where the existing lakes are small and currently less numerous but where new lakes develop and the

existing ones show rapid (over 100 %) increase (Fig. 8 a, c). The mean volume is 0.33x10$^6$ m$^3$ and this is

close to the mean volume of 32 Type 1 and Type 2 lakes (whose bathymetry was measured; Sect.3.4) of

0.47x10$^6$ m$^3$.

Frequency distributions, mean and median values of the selected morphometric parameters of the

modelled overdeepenings were compared to those of 134 Type 1 and Type 2 lakes with areas in excess of

11,000 m$^2$ as in 2014 (Fig. 11). The frequency distributions of areas of the modelled overdeepenings and

existing lakes are very similar, however, both mean and median areas of the overdeepenings are smaller

than those of the lakes (Fig. 11a). The majority of the existing lakes have maximum lengths and width of

200-400 m and 100-200 m respectively and their width-to-length ratio (elongation) peaks at 0.6-0.8

pointing at lower eccentricity in line with the prevalence of the cirque glaciers in the region currently and

in the past (Vilesov et al., 2013). While frequency distribution of the maximum length values is replicated

by the model, the majority of the modelled overdeepenings tend to be narrow probably because most

overdeepenings are small. Thus elongation of 44% of the overdeepenings is within 0.4-0.5 range (Fig. 11

e) while areas of 38% of the overdeepenings do not exceed that of two model grid cells.

The locations and characteristics of overdeepenings were analysed with regard to the lakes formed within

the assessment period in the de-glacierzed area. All actual lakes, including those smaller than 11,000 m$^2$,

were considered.  Overall, 66 actual lakes (56 new and 10 whose formation began earlier) were identified





within this domain ranging in size between 960 m$^2$ and 44,260 m$^2$. Within the same area, 148

overdeepenings were identified (116 within the fully de-glacierized area). Positions of 44 overdeepenings

coincided with positions of the actual lakes, i.e. 67 % match. The mean and median areas of these

overdeepenings were 60,600 m$^2$ and 39,400 m$^2$ respectively.

Not all overdeepenings will become lakes as those that are smaller and shallow will be filled by sediment

(Linsbauer et al., 2012; 2016). In this study, 72 overdeepenings were identified as 'false positives', e.g.

no lakes formed in place of the identified overdeepenings following the complete retreat of glaciers by

2014. The remaining overdeepenings were within the partially de-glacierized area where lakes can

potentially develop. Most of the 'false positive' overdeepenings were shallow (maximum depth less than

25 m) and small with individual areas less than 0.05 km$^2$ and mean and median areas of 18,900 m$^2$ and

16,875 m$^2$. Areas of 50 % of the 'false positive' overdeepenings were at the imposed threshold of two

495    grid cells.

Fig. 10 (b) shows an overdeepening with the largest area in the data set is positioned at the top of a cascade

of lakes in the River Bien basin identified as in danger of GLOF. In the worst-case scenario, filled to the

brim, this overdeepening may contain 9.4x10$^6$ m$^3$ of water which exceeds the volume of Lake Kapkan,

the only artificially lowered lake, approximately by the factor of three. At the exact site of this

500    overdeepening, modelled using glacier mask from the year 2000, Lake 22 formed and increased rapidly

between 2002 and 2014 (Fig. 10b; Table S3; Sect. 4.3) and its area can potentially increase by an order of

magnitude in the future. In the Aksu basin, the formation and growth of Lakes 9 and 13 (by 57 % and 123

% respectively) coincided with the modelled overdeepenings, however, modelling suggests that potential

for the increase of Lake 13 is limited (Fig. 9). Similarly, limited expansion is projected for Lake Kapkan

505    (N 25) although multiple overdeepenings with individual areas between 40,000 m$^2$ and 70,000 m$^2$ were

identified in its vicinity suggesting that new lakes may form (not shown). Lake Akkol (N 43) is not

expected to increase according to the results of modelling and indeed, the lake area did not change between

2002 and 2014 (Table S3).



## 5. Discussion

### 5.1. The inventory and evolution of lakes

This study assessed changes in approximately 600 unmanaged lakes in the Djungarskiy Alatau between 2002 and 2014 with regard to various types and size classes of lakes and characteristics of the catchments. The overall number of lakes and their combined area increased between 2002 and 2014 by 6.2 % and 6.9 % representing growth rates of 0.51 % $a^{-1}$ and 0.55 % $a^{-1}$ respectively. These results are comparable to the increase of 7 % (0.7 % $a^{-1}$) in lake count and 8.4 % (0.84 % $a^{-1}$) in lake area presented by Wang et al. (2013) for a wider region of the northern Tien Shan for the 2000-2010 period. The difference between the two studies is close to the uncertainty of measurements and can be attributed to the difference between methodologies (automated versus manual mapping) and lake size distributions in the samples. Narama et al. (2010b) reported smaller increase in area of 36 lakes in the Terskey Alatoo in the 1998-2008 period attributing it to the comparatively slow glacier retreat in the region.

The overall increase in both count and area is relatively small. This is partly because the sample is dominated by small lakes for which uncertainty is higher and we consider changes in 68 % of all lakes to be within the uncertainty of measurements. In this study, 7 %, 17 % and 24 % of lakes were smaller than the thresholds of 2,000 $m^2$, 3,600 $m^2$ and 4,500 $m^2$ applied by Wang et al. (2013), Gardelle et al. (2011) and Li and Sheng (2012) respectively in their assessments of changes in lake areas in the Tien Shan, Hindukush and Himalayas. Another reason is a dynamic behaviour of the lakes. In contrast to changes in area of glaciers, which retreated in the region during the 2000s (Severskiy et al., 2016; Table S1), both the expansion and formation of lakes and their drainage were registered and the average statistics conceals significant variations.

In the sub-sample of lakes whereby changes exceeded the uncertainty of measurements, the increase in lake count and area dominated over reduction. Areas of 116 lakes increased and 69 new lakes formed



while areas of 42 lakes decreased and 32 lakes drained completely (Table 2; Fig. 6). Similar diversity in

lake evolution and rates between increasing and shrinking numbers of lakes was reported by Li and Sheng

(2012) for the Himalayas. The largest increase in lake count characterised Type 1 (contact) lakes (Table

4) and similar results were reported by Wang et al. (2012) and Gardelle et al. (2011) for the Hindukush

and the Himalayas. The Type 2 lakes, developing on the young moraines, were the second largest group

exhibiting growth, however, the most active reduction in lake area was also registered in this sub-set

usually following glacier retreat and separation between lakes and glacier tongues.

In comparison with other regions of the Tien Shan, the number of lakes was increasing marginally faster

in the Djungarskiy Alatau than in the western and central Tien Shan where the abundance of lakes was

growing at a rate of 0.4 % $a^{-1}$ but slower than in the eastern Tien Shan where the increase in the number

of lakes reached 1.7 % $a^{-1}$ between 2000 and 2010 (Wang et al., 2013).  The increase in combined area

was marginally higher than in the western Tien Shan (0.4 % $a^{-1}$) but lower than in the central (0.84 % $a^{-1}$)

and eastern (1.49 % $a^{-1}$) Tien Shan (Wang et al., 2013).  The growing numbers and areas of lakes in the

glacierized regions contrasted strongly with evolution of lakes on the adjacent plains where the large-scale

atmospheric circulation controls (and subsequently variability in temperature and precipitation) are the

same but glacier melt does not contribute to lake nourishment. Thus during 1987-2010, on the plains of

Mongolia, the number of lakes with individual areas in the range of 1-10 $km^2$ and their combined area

decreased by 10.5 % and 28 % respectively following a strong increase in evaporation (Tao et al., 2015).

**5.2. The observed climate and glacier change and lake evolution**

The evolution of lakes resulted from a combination of the following factors (i) climatic changes including

changes in temperature and difference between precipitation and potential evaporation (PET); (ii) glacier

melt nourishing lakes; (iii) melt of buried ice and permafrost affecting both lake nourishment and



drainage; (iv) local topography and stability of moraines supporting the lakes. Contributions of these factors vary between lake types, temporally and spatially.

There are no high-altitude meteorological stations with continuous records in the study area, however, significant positive temperature trends were identified at the stations located between approximately 600

m and 1700 m a.s.l. in all seasons including the summer ablation season. The JJA temperatures were increasing at a rate of 0.18ºC per decade between 1960 and 2014 at the regional stations (Fig. 1) with linear trends, significant at 0.05 confidence levels, explaining approximately 20 % of the variance in the data set. Positive trends in summer temperatures are observed across the Tien Shan (Kutuzov and Shahgedanova, 2009; Narama et al., 2010a; Wang et al., 2013). Vilesov et al. (2013), estimated potential

evapotranspiration (PET) for the Mynzhylki meteorological station (3010 m a.s.l.) in the Zailiiskiy Alatau concluding that while the long-term record exhibits a statistically significant positive trend consistent with temperature increase, the change in PET between the 1990s and 2000-2010 was small (about 5 % of the annual mean). Trends in precipitation vary between the mountainous regions of Central Asia (Kutuzov and Shahgedanova, 2009; Narama et al., 2010a; Wang et al., 2013) but in the foothills of the Djungarskiy

Alatau as well as at the high-elevation stations of the Zailiiskiy Alatau neither annual nor summer precipitation exhibited statistically significant trends both between 1960 and 2014 and within the assessment period. We suggest that changes in effective precipitation (difference between precipitation and PET) were not the main control over lake evolution in the Djungarskiy Alatau in contrast to the adjacent plains (Tao et al., 2015).

Glacier mass balance is a useful predictor of lake evolution (Gardelle et al., 2011; Wang et al., 2015). While no continuous records exist for the assessment period in the Djungarskiy Alatau, mass balance records from the Tuyuksu glacier in the Zailiiskiy Alatau show predominantly negative annual and strongly negative cumulative mass balance (Severskiy et al., 2016). The mass balance data derived from GRACE and ICESat (Farinotti et al., 2015) show that values of negative mass balance in the Djungarskiy

Alatau are close to those observed in the Zailiiskiy Alatau and indicate mass loss of approximately -0.5 x





$10^3$ kg m$^{-2}$ a$^{-1}$. As a result, between 2002 and 2014 glaciers lost between 5 % and 30 % of their combined

area (Table S1) contributing to the expansion of particularly contact Type 1 lakes, 73 % of which increased

and 51 new lakes formed (Table 4).

Degradation of buried ice and permafrost controls the growth of morainic lakes and their discharge often

as GLOF events resulting from buried ice erosion and moraine dam instability in the Tien Shan (Jansky

et al., 2010; Evans and Delaney, 2015). By contrast, Severskiy et al. (2013) noted that many morainic

lakes that existed in the 1970s in the Tien Shan gradually disappeared in the 21$^{st}$ century and attributed

this trend to thawing of permafrost (Severskiy, 2009). Permafrost originally formed an impermeable layer

on which lakes developed but as it thawed, the active formation of underground drainage channels enabled

gradual drainage of the lakes. Field surveys revealed the widespread presence of buried ice in the

contemporary moraines in the Djungarskiy Alatau where Type 2 lakes develop but not in the LIA and

older moraines where Type 3 lakes occur (Vilesov et al., 2013). In addition to the effect of glacier

nourishment, this difference might explain the relatively small changes in Type 3 lakes over the relatively

short period of time and more dynamic behaviour of Type 2 lakes (Table 4). The effect of melting

permafrost on dam stability can be monitored using multi-temporal high-resolution satellite imagery and

DEM (Fujita et al., 2008; Narama et al., 2010b; Bolch et al., 2011) and this analysis will be performed in

the future.

**5.5. Risks of lake outbursts and modelling overdeepenings in glacier beds for hazard management**

GLOF events peaked in the Djungarskiy and neighbouring Zailiiskiy Alatau between 1975 and 1984 (Fig.

12) when strong positive JJA temperature anomalies and enhanced glacier melt were observed. The

frequency of GLOF was lower in the late 1980s when glacier retreat was slower and when management

of lakes was introduced in the more densely populated Zailiiskiy Alatau (Severskiy et al., 2013). Despite

these efforts, two GLOF events occurred in 2014 and 2015 in the Zailiiskiy Alatau following strongly



positive JJA temperature anomalies of 2-3°C (reaching 6°C in July 2015) recorded at the Tuyksu

meteorological station (3440 m a.s.l.). In August 2014, periodic artificial lowering of Lake Kapkan

commenced in the Djungarskiy Alatau to prevent its outburst.

Fifty lakes, whose outburst can potentially pose threat to existing infrastructure, were identified (Table

S3) as a first step in the assessment of GLOF risk. Most are large enough to cause damage in case of an

outburst of a single lake, however, smaller lakes can pose danger due to their potential to trigger

overtopping of lakes located downstream. Many lakes exhibited strong growth within the assessment

period (Table 4; S3) and are projected to increase in the future. The use of the most stringent criteria

derived from the previous GLOF events, literature and empirical models (e.g. peak discharge resulting

from the worst case scenarios) implies that anticipated floods are low probability events and further

justification through physical modelling of floods and mudflows and field surveys, focusing on the state

of lake dams, are required. An example, where further modelling of potential floods is needed to constrain

uncertainty and assist planning, is the basin of the River Bien where a rapid increase of newly formed

lakes is observed, comparatively close to the existing infrastructure, and their further increase is projected

potentially resulting in a formation of one of the largest lakes in the region (Fig. 10b) but where the extent

of flood propagation is uncertain.

Modelling future evolution of dangerous lakes, based on the identification of overdeepenings in the

modelled glacier beds, may be used to inform hazard management and planning. Previous studies have

shown that while simulation of locations of overdeepenings is robust (Frey et al., 2014; Linsbauer et al.,

2012; 2016), their morphometry is subject to considerable uncertainty (Haeberli et al., 2016). A

comparison of morphometry of the overdeepenings in the Djungarskiy Alatau with that of the existing

lakes (Fig. 11) shows that their parameters, including mean statistics and frequency distributions, are close

providing a degree of confidence in the simulations. The frequency of slightly larger areas (0.05-0.1 km$^2$)

is higher in the modelled sample but both samples show that very small lakes (0.01-0.05 km$^2$) prevail.

Narrow overdeepenings, under 100 m in width, are most frequent (44 %) while the majority of actual

lakes (57 %) are 100-200 m wide (Fig. 11d). This discrepancy is likely to be an artefact of a large number

of small (two grid cell) overdeepenings in the sample. As a result, oval-shaped overdeepenings with

elongation of 0.4-0.5 prevail while actual lakes tend to have more circular form.

A comparison with the morphometries modelled for the Swiss Alps, Himalaya-Karakoram region and

Peruvian Cordillera Blanca (Haeberli et al., 2016) shows areas of the overdeepenings in the Djungarskiy

Alatau are smaller than in all three regions in line with smaller size of glaciers. Maximum depth statistics

is comparable with that for the Himalaya-Karakoram and the Cordillera Blanca. Shallow overdeepenings,

with maximum depth less than 50 m dominate in all regions and in the study area, maximum depth values

are less than 40 m for 95 % of all identified overdeepenings.

A correct simulation of locations of overdeepenings is important in application to hazard management.

Our comparison of locations of the contact lakes, formed since 2000, with modelled overdeepenings

showed that locations of 67 % of the lakes were simulated correctly although the number of 'false

positives' was high.  Following the analysis of morphometries of the 'false positives' and a comparison

of morphometries of the existing lakes and the overdeepenings (e.g. width and elongation, Fig. 11), we

recommend that a threshold of 20,000 m$^2$ is applied to identify overdeepenings which are likely to provide

sites for the development of lakes as 85 % of the actual lakes formed in the overdeepenings with areas

exceeding this threshold.

A promising outcome of this investigation is a good agreement between the simulated areas of

overdeepenings and the observed evolution of the lakes. For example, areas of two lakes, the Kapkan

(prior to its artificial lowering) in the Khorgos basin and Akkol in the Sarkand basin, currently considered

by KSAMP as the most dangerous in the region, did not increase between 2002 and 2014 and modelling

suggests that they already have filled overdeepenings in the terrain. However, a strong growth of Lake 22

in the Bien catchment (Table S3) occurred in place of the largest of the identified overdeepenings (Fig.

10b). These and other (e.g. formation of multiple lakes in larger overdeepenings) examples suggest that

GlabTop modelling approach is a useful tool for planning and hazard management.


## 6. Conclusions

An increase in abundance and combined area of mountain lakes was observed in the Djungarskiy Alatau

between 2002 and 2014. The overall change was moderate at approximately 6 %, however, the small

overall increase resulted from a very dynamic behaviour of lakes which involved both growth and

drainage of lakes over time. Two categories of lakes, those developing on the young moraines and contact

lakes, exhibited the strongest growth which agreed well with projected areas of overdeepenings in the

subglacial topography simulated using GlabTop2 model. Fifty existing lakes were identified as potentially

dangerous. In the future, the significance of hazard potential of these lakes will be assessed trough further

investigation of lake dams and surrounding slopes using high-resolution remote sensing and field studies

and physically-based flow models.

Over 500 overdeepenings in the glacier beds were identified using GlabTop2 model. A comparison of

locations of the modelled overdeepenings and actual lakes formed within the recently de-glacierized area

shows that GlabTop2 can serve as a useful tool in hazard management in the region. The application of

area threshold of 20,000 m$^2$ to the identified overdeepenings may help to reduce the number of

overdeepenings which are unlikely to provide sites for the development of lakes. The uncertainties of

modelling and the ongoing development of the region imply that the observed and modelled evolution of

the lakes should be regularly re-assessed against each other and against changes in land use and

infrastructure to inform hazard management and planning.

**Author Contribution**

M. Shahgedanova designed the study, contributed to data analyses (identification of overdeepenings;

statistical analysis) and has written the text; V. Kapitsa carried out lake mapping and identification of

potentially dangerous lakes, contributed to statistical analysis; H. Machguth ran GlabTop2 model and




contributed to writing the text; I. Severskiy and A. Medeu contributed to study design and discussion of

the results.

**Acknowledgements**

This work was conducted as a part of the project "Climate Change, Water Resources and Food Security
in Kazakhstan" funded by Newton - al-Farabi Fund (grant No 172722855). We thank Kazakhstan State

Agency for Mudflow Protection (KSAMP) and personally Murat Kasenov for sharing their knowledge

with us and for incorporating our results in their work.

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



Table 1. Characteristics of the Landsat images used in this study. The July 2014 scene covered area containing 15 lakes (2 % of all lakes as in 2014).

| Satellite | Sensor | Path / row | Scene ID number | Acquisition date |
|---|---|---|---|---|
| Landsat 7 | ETM+ | 148/29 | LE71480292002237SGS00 | 25.08.2002 |
|  |  | 147/29 | LE71470292002230SGS00 | 18.08.2002 |
|  |  | 147/28 | LE71470282002214SGS00 | 02.08.2002 |
| Landsat 8 | OLI | 148/29 | LC81480292014214LGN00 | 02.08.2014 |
|  |  | 147/29 | LC81470292014223LGN00 | 11.08.2014 |
|  |  | 147/28 | LC81470292014191LGN00 | 10.07.2014 |





Table 2. Distribution of lakes by type in 2002 and 2014. Lake areas (combined and mean) are in km$^2$;

elevation in m a.s.l. Elevation given in parentheses refers to 2014.

| Lake type | 2002 | | | 2014 | | | Elevation |
|---|---|---|---|---|---|---|---|
| | No | Total | Mean | No | Total | Mean | |
| Type 1 | 107 | 2.39±0.13 | 0.022 | 118 | 2.62±0.15 | 0.022 | 3020-3660 (3690) |
| Type 2 | 203 | 2.93±0.19 | 0.014 | 234 | 3.8±0.23 | 0.016 | 2910-3660 |
| Type 3 | 267 | 8.63±0.43 | 0.032 | 263 | 8.64±0.43 | 0.033 | 2470-3590 |
| Type 4 | 22 | 2.31±0.1 | 0.11 | 21 | 2.27±0.09 | 0.11 | 2220-3220 |
| Total / mean | 599 | 16.26±0.85 | 0.02 | 636 | 17.35±0.9 | 0.03 | 2220-3690 |




Table 3. Changes in number and the combined areas of lakes.

| Change 2002-2014 | No | Combined area (km$^2$) | | Change in combined area | |
|---|---|---|---|---|---|
| | | 2002 | 2014 | km$^2$ | % |
| Increased | 116 | 1.43±0.097 | 2.41±0.15 | 0.98±0.26 | 69±17 |
| Decreased | 42 | 0.68±0.042 | 0.46±0.03 | 0.22±0.08 | 32±17.4 |
| Change within uncertainty | 409 | 13.91±0.7 | 14.06±0.71 | 0.15±1.3 | 1±15.1 |
| New | 69 | - | 0.44±0.03 | 0.44±0.18 | 100±15.3 |
| Drained completely | 32 | 0.25±0.02 | - | 0.25±0.09 | 100±15.6 |





Table 4. Changes in number and combined area of lakes of different types. For each category, the upper and the lower lines show the absolute (km$^2$) and relative (%) changes respectively. Lakes classified as Type 1 in 2002 and Type 2 in 2014 are shown separately as 'changed type' category.

| Lake type / change | Type 1 | | Type 2 | | Type 3 | | Type 4 | | Changed type | |
|---|---|---|---|---|---|---|---|---|---|---|
| | N | km$^2$ / % | N | km$^2$ / % | N | km$^2$ / % | N | km$^2$ / % | N | km$^2$ / % |
| Increased | 57 | 0.74± 0.17 75±15.2 | 29 | 0.086±0.036 47.3±18.8 | 8 | 0.024±0.013 31.9±21.2 | 2 | 0.012±0.004 37.8±23 | 20 | 0.1±0.029 78.6±16.9 |
| Decreased | - | - | 25 | 0.13±0.041 35.3±16.7 | 7 | 0.014±0.01 23.4±18.3 | 5 | 0.058±0.02 28.2±13.2 | 5 | 0.017±0.004 41.9±21.9 |
| Changed within uncertainty | 10 | 0.03±0.058 5.7±13.8 | 127 | 0.099±0.28 4.05±16.6 | 248 | 0.14±0.95 2.2±14.7 | 14 | 0.058±0.22 2.9±12.1 | 10 | 0.034±0.068 5.3±13.5 |
| New | 51 | 0.305±0.13 | 18 | 0.13±0.051 | - | - | - | - | - | - |
| Drained | 5 | 0.032±0.009 | 22 | 0.18±0.015 | 4 | 0.023±0.005 | 1 | 0.021±0.005 | - | - |





**Figure captions**

Fig. 1. Study area. DGLOF symbols show lakes identified as potentially in danger of outburst.

Meteorological stations: 1 – Sarkand (764 m a.s.l); 2 – Lepsy (1012 m a.s.l.); 3 – Jarkent (643 m a.s.l.).

Fig. 2.  Temperature and precipitation climatology for Lepsy meteorological station (45.32º N; 80.37º E;

1012 m a.s.l.), 1960-2014. Location of the station is shown in Fig. 1.

Fig. 3. An example of the upper section of a cascade (vertical sequence) of lakes in the Aksu River basin

(an oblique aerial photograph by M. Kasenov; July 2014). Fig. 9 (a) shows a Landsat scene depicting the

cascade.

Fig. 4. Examples of lakes of different types: Landsat imagery (2014) and oblique aerial photographs (V.

Kapitsa; M. Kasenov; July 2014): (a) Type 1 (contact lakes); (b) Type 2 (proglacial lakes on 20-21 C

moraines); (c) Type 3 (proglacial lakes on LIA and older moraines); (d) Type 4 (rock-dammed lake).

Capital letters refer to the corresponding lakes shown on the pairs of satellite images and photographs.

Fig. 5. Relationship between lake area and volume in the Zailiiskiy and Djungarskiy Alatau.

Fig. 6. (a) Number of lakes, (b) their combined areas, relative change in lake areas averaged over size

category and number of lakes whose areas (c) increased and (d) decreased. X-axes show size categories

(km$^2$). The vertical bars represent uncertainty of measurements. Note that different scales are used in (c)

and (d).

Fig. 7. Lake count (numbers) and combined area (bars) by elevation (m a.s.l.). The black bars represent

uncertainty of area measurements.

Fig. 8. Spatial distribution of lakes which (a) increased in size, (b) decreased in size, (c) newly formed

and (d) drained completely.

Fig. 9. (a) Cascade of lakes in the Aksu basin (Landsat scene from 2014) and (b) modelled overdeepenings

using glacier mask from 2000 and actual lakes as in 2014.

Fig. 10. (a) Cascades of lakes in the Bien River basin (Landsat scene from 2014) and (b) modelled

overdeepenings using glacier mask from 2000 and actual lakes as in 2014.



Fig. 11. Histograms of the selected parameters of 513 modelled overdeepenings and 134 actual Type 1
and Type 2 lakes with areas in excess of 11,000 m$^2$ as in 2014: (a) Area (km$^2$); (b) maximum depth (m);

(c) maximum length (m); (d) maximum width (m); (e) elongation (width to length ratio).

Fig. 12. GLOF events, attributed to increase in air temperature and melt (as opposing to rainfall), which

resulted in the recorded damage to housing and infrastructure in the Zailiiskiy and Djungarskiy Alatau.

The data before 1985 are from Popov (1986); data for the later years were provided by KSAMP.





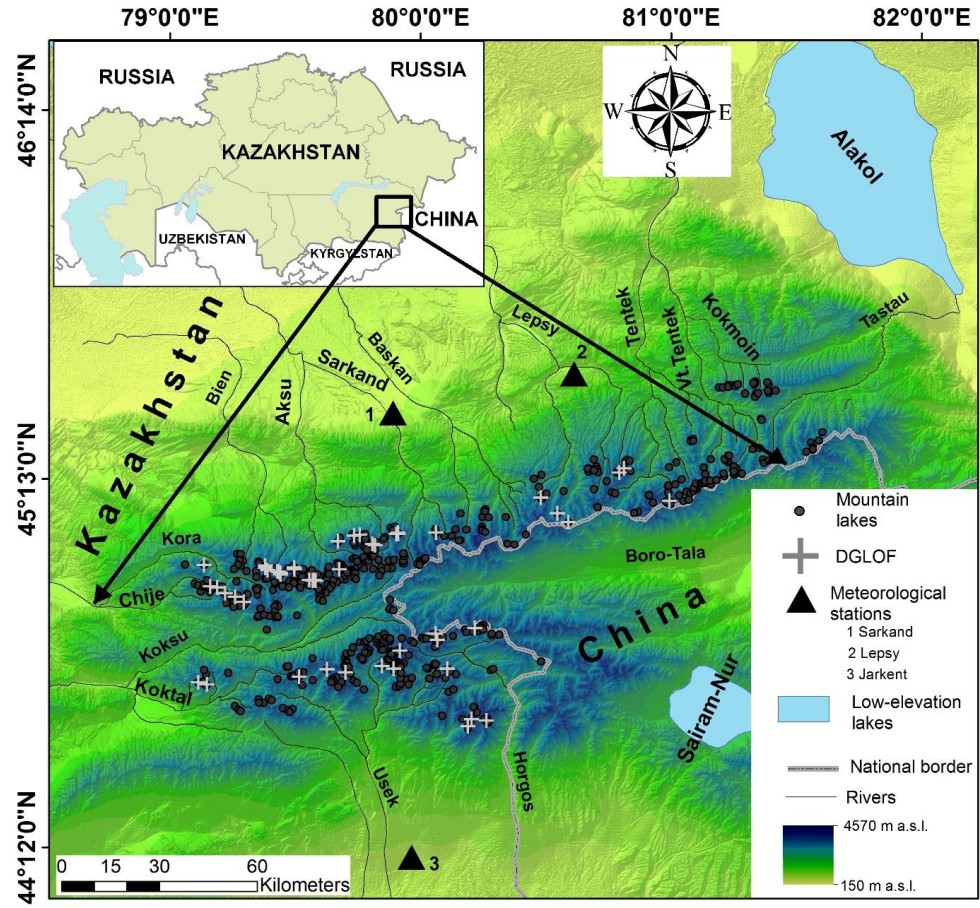


Fig. 1.



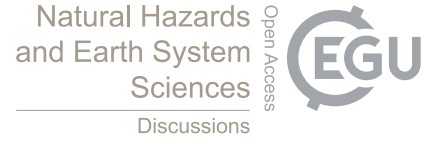

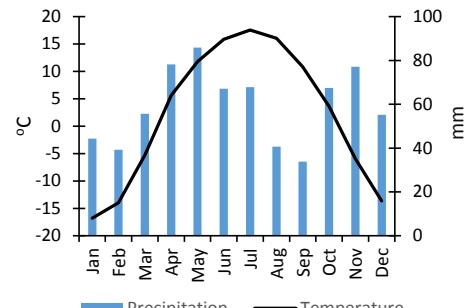

Fig. 2.






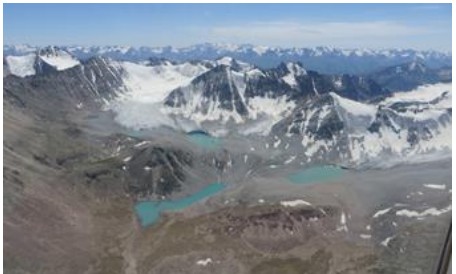

Fig. 3.




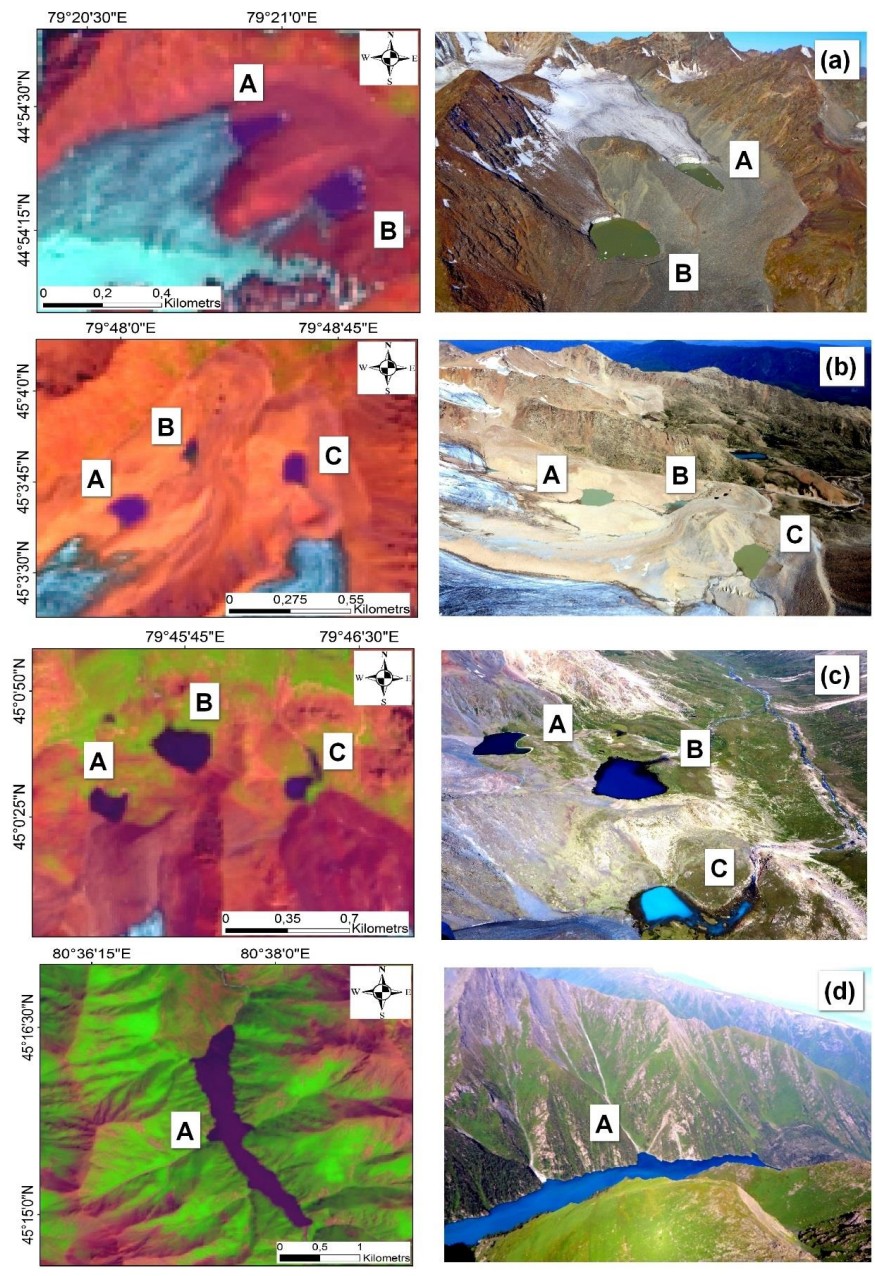

Fig. 4.





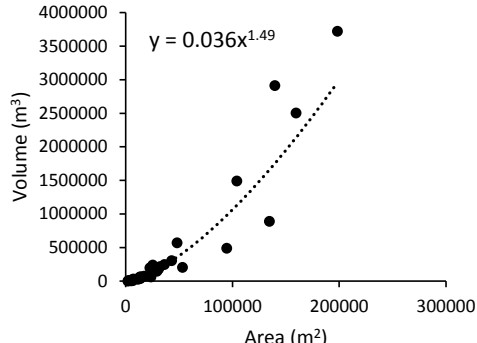

Fig. 5.






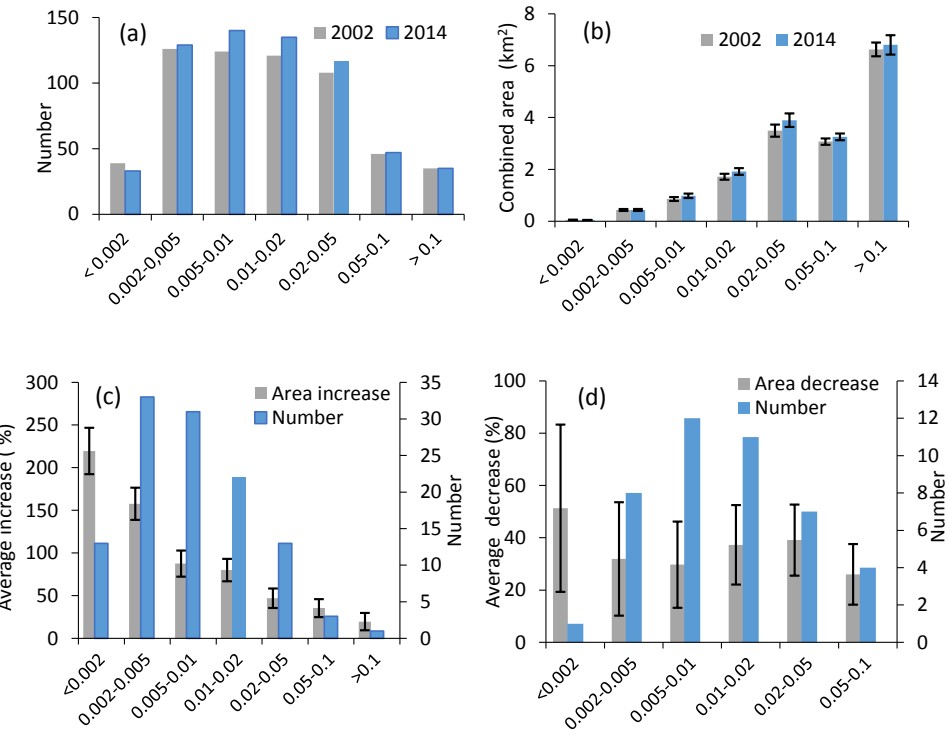

Fig. 6.




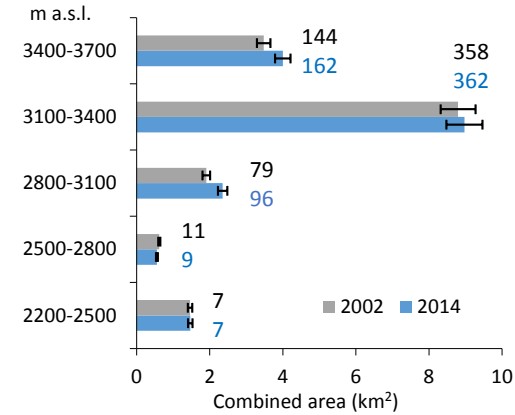


Fig. 7.






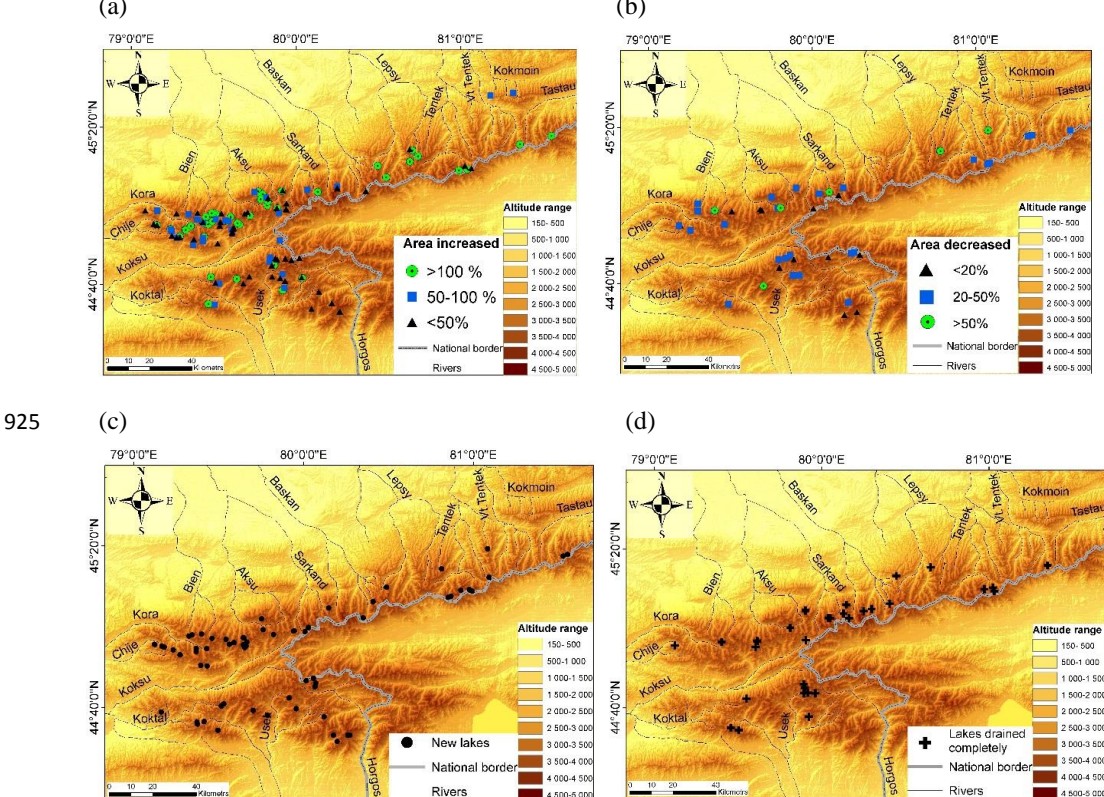

Fig. 8.




(a)

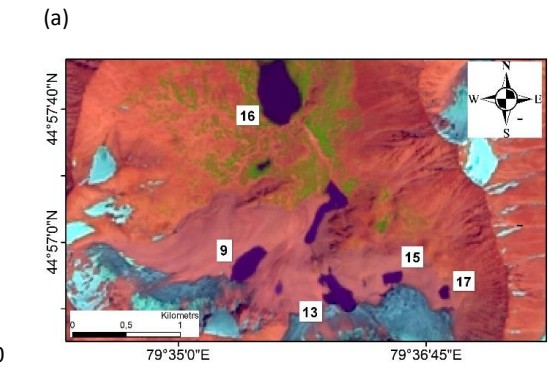


(b)

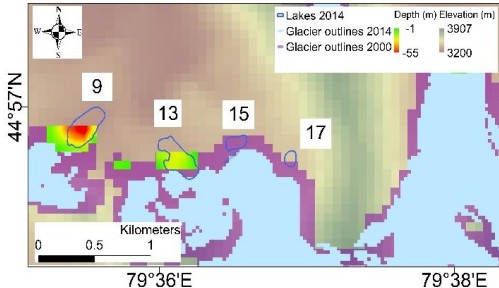

Fig. 9.



(a)

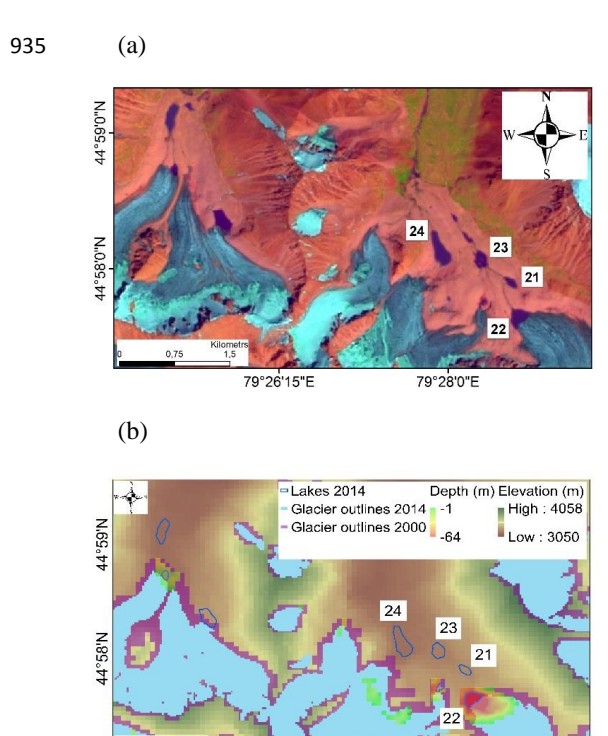

(b)

Fig. 10.




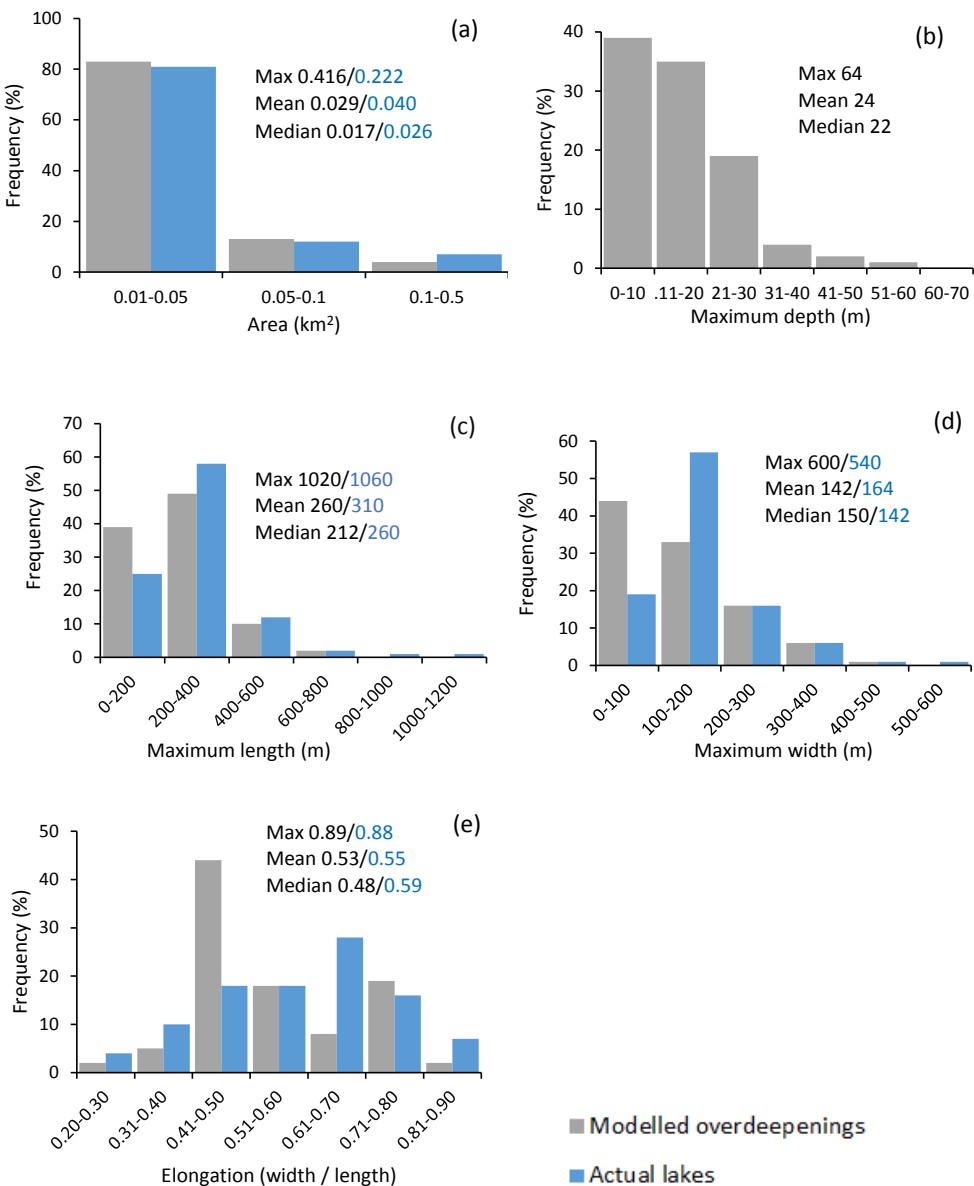


Fig. 11.





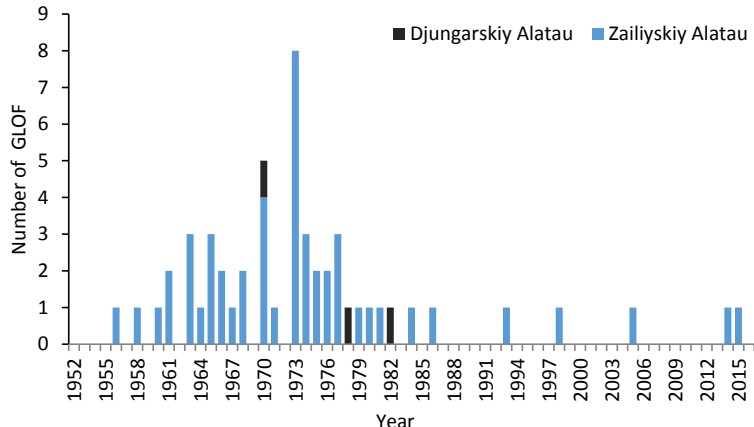

Fig. 12.
