# Peer review of "Assessment of Evolution and Risks of Glacier Lake Outbursts in the Djungarskiy Alatau, Central Asia, using Landsat Imagery and Glacier Bed Topography Modelling"

_Natural Hazards and Earth System Sciences, 2017_

## Referee Comment (RC1) · Anonymous Referee #1 · 3 Jul 2017

This is a solid regional-scale assessment of potential dangers from the outbursts of mountain lakes, using repeat satellite images, and related glacier bed modelling to identify overdeepenings. Good work! I recommend acceptance after minor revisions, some of which I think are important to consider, though.

Important comments:

Line 17 (L17) and at several places the 6.2% and 6.6% lake growth was confusing to

me, because its significance level is unclear. Better focus (as done later, L365) on the number of lakes that showed significant growth, or similar. Note, changes without error bars, or some other indication of uncertainty are worthless. And I don't understand the mean value of uncertainty L200 and its meaning in this context.

L273: threshold for what? L275: how good is the ASTER GDEM in particular over steep terrain, and how reliable is thus this slope threshold computation?

L323: can you explain why you use the ASTER GDEM for slope assessments and SRTM for glacier bed estimation?

Other comments:

- the paper needs some grammar corrections, but I believe this is done by Copernicus.

- the title is quite long. Try to shorten?

L18, L132: contact with what?

L215: are their supraglacial lakes in the region? Is none of the ice contact-lakes ice-dammed? Sure?

L299: sudden and complete drainage?

END

---

## Referee Comment (RC2) · Anonymous Referee #2 · 13 Aug 2017

General comments. The paper provides comprehensive assessment of evolution of mountain lakes in the Djungarskiy Alatau based on comparison of multi-temporal Landsat imagery, with identification of 50 potential GLOF hazardous lakes. GlabTop2 approach was tested on data obtained and applied to predict appearance of new lakes following deglaciation. It is substantive study with good regional coverage and I recommend accepting this paper after minor revisions.

[Figure]

Specific comments. Line 16. "In 2002 and 2014, 599 lakes with a combined area of 16.26±0.85 km2 and 636 lakes with a combined area of 17.35±0.92 km2 respectively were identified". In this context, it looks like the area change lies within the uncertainty in measurements.

Lines 55-64. It seems quite appropriate to mention here (and discuss somewhere in the text) the new glacial lake inventory with assessment of their outburst potential for Central Asia: Petrov M. A., Sabitov T. Y., Tomashevskaya I.G., and others (2017). Glacial lake inventory and lake outburst potential in Uzbekistan. Science of the Total Environment, 2017 Aug 15; 592: 228-242. doi: 10.1016/j.scitotenv.2017.03.068. Epub 2017 Mar 17.

Lines 171-172. Channels 7, 4, 2 and panchromatic channel 8 of Landsat 7 are not the same as of Landsat 8. Their numbers are different and, in some cases, their wavelength bands also

Lines 173-174. The threshold of digitization should 675 m2 but not 700 m2 (taking three 15×15 m pixels as lowest limit of lake identification).

Line 346. The absolute vertical accuracy of ASTER GDEM2 of 17 m is given from (Meyer et al., 2011). But what is the vertical accuracy of ASTER GDEM2 relative to ice-free areas on topographic maps of Djungarskiy Alatau?

Line 666. I suggest to specify in the Conclusions : in which areas (basins) of the Djungarskiy Alatau the formation of a large number of new lakes is expected.

Technical corrections.

* In general, the text of the paper can be slightly reduced without affecting its content.

Lines 238-240. This phrase looks too heavy: "While increase in lake area is a factor making lakes outburst more likely (Bolch et al., 2011), no change or reduction in lake area is not a guarantee that outburst will not occur because 240 potential thawing of ice contained within the morainic dam (Jansky et al., 2010; Herget et al., 2013; Evans

and Delaney, 2015) or blockage of channels within the dam (Narama et al., 2010b) can lead to its breach in a short period of time".

Lines 443 and 444. The paper (Farinotti et al., in Press ) is already published: Farinotti et al., 2017

Line 724. The paper is already published: Farinotti, D., Brinkerhoff, D. J., Clarke, G. K. C., Fürst, J. J., Frey, H., Gantayat, P., Gillet-Chaulet, F., Girard, C., Huss, M., Leclercq, P. W., Linsbauer, A., Machguth, H., Martin, C., Maussion, F., Morlighem, M., Mosbeux, C., Pandit, A., Portmann, A., Rabatel, A., Ramsankaran, R., Reerink, T. J., Sanchez, O., Stentoft, P. A., Singh Kumari, S., van Pelt, W. J. J., Anderson, B., Benham, T., Binder, D., Dowdeswell, J. A., Fischer, A., Helfricht, K., Kutuzov, S., Lavrentiev, I., McNabb, R., Gudmundsson, G. H., Li, H., and Andreassen, L. M.: How accurate are estimates of glacier ice thickness? Results from ITMIX, the Ice Thickness Models Intercomparison eXperiment, The Cryosphere, 11, 949-970, https://doi.org/10.5194/tc-11-949-2017, 2017.

Lines 869 and 900. Abbreviation "DGLOF" in caption of Figure 1 and in legend on this figure is not explained. The rest of the text uses the "GLOF" abbreviation. What is the difference? Unify. . .

Lines 930 and 935. It would be good to increase the readability of lines indicating "Lakes 2014" on Figures 9b and 10b.

---

## Author Comment (AC1) · 2 Sep 2017

**Response to the comments made by Reviewers 1 and 2**

We are grateful to both anonymous reviewers for their comments which we found very helpful. We have made all the corrections and implemented all suggestion. The detailed answers are given below. The revised version of the manuscript is ready for submission.

**Reviewer 1**

1.1. Comment: Line 17 and at several places the 6.2% and 6.6% lake growth was confusing to me, because its significance level is unclear. Better focus (as done later, L365) on the number of lakes that showed significant growth, or similar. Note, changes without error bars, or some other indication of uncertainty are worthless. And I don't understand the mean value of uncertainty L200 and its meaning in this context.

Line 17: Reference to 6.6% increase was removed: "The combined areas were $16.26\pm0.85$ km$^2$ to $17.35\pm0.92$ km$^2$ respectively and the overall change was within the uncertainty of measurements."

We agree that the mean value of uncertainty in Line 200 is misleading. This phrase has been removed. Uncertainty values have been added throughout the text.

1.2. Comment: L273: threshold for what?

This has been clarified: "Slopes steeper than 45º are considered as particularly dangerous in this regard (Alean, 1985; Bolch et al., 2011; Cook et al., 2016)."

1.3. L275: how good is the ASTER GDEM in particular over steep terrain, and how reliable is thus this slope threshold computation?

Firstly, see response to comment 1.4: SRTM3 GDEM was used in the original version.

Secondly, we have a limited number of ground-based geodetic measurements in the study region to quantify the uncertainty of DEM derived from satellite data. We have added RMSE values for both ASTER and SRTM based on several ground control points (GCP); see answers to comment 2.5. However, only a small number of GCP were available and we used a comparison between the two DEM to characterise uncertainty. The following text has been added:

Sections 3.1: "The void-filled SRTM3 GDEM (https://lta.cr.usgs.gov/SRTM1Arc) and ASTER GDEM2 (https://asterweb.jpl.nasa.gov/gdem.asp) with 30 m resolution were used to derive data on slope angles. SRTM3 DEM was used to derive elevations of the lakes. A reliable GDEM of the study area is essential for the assessment of thresholds for mass movements in

the vicinity of lakes and potential debris flow pathways. Previously, both GDEM were shown to be suitable for assessments of slope angles and elevations in the northern Tien Shan with limitations regarding smaller features such as lateral moraines and deep gorges (Bolch et al., 2011). In the absence of ground-based measurements in the study area, a comparison between slope angles derived from ASTER and SRTM was used to characterise uncertainty."

Section 3.4: "Slope values derived from ASTER GDEM exceeded those derived from SRTM: within the 500 m distance from Type 1 and 2 lakes, mean slope values and standard deviations were $16.9\pm4.8^o$ and $21.3\pm4.0^o$ for SRTM and ASTER respectively."

Section 5.5: "Results of modelling of outburst paths and mass movements depend on the quality of GDEM. The average difference between SRTM and ASTER GDEM was approximately $4^o$ for the slopes surrounding lakes with steeper slopes derived from ASTER GDEM. A detailed comparison of two GDEM and their validation will require extensive ground truth data which is currently unavailable. Due to the stringent criteria applied to the selection of dangerous lakes (i.e. distance to the infrastructure and average incline along the potential flood path line), the DEM uncertainty does not affect the 'first-order' identification of dangerous lakes presented here. However, a more accurate DEM and evaluation of its quality may be required for modelling of debris flow propagation."

.

1.4. Comment: L323: can you explain why you use the ASTER GDEM for slope assessments and SRTM for glacier bed estimation?

This is an unfortunate mistake which propagated through the text: SRTM3 GDEM was used for slope assessment and to calculate elevation bands in which lakes are positioned. The text has been corrected in Section 3.1 and throughout the text: "SRTM GDEM (https://lta.cr.usgs.gov/SRTM1Arc) and ASTER GDEM2 (https://asterweb.jpl.nasa.gov/gdem.asp) with 30 m resolution were used to derive data on slope angles while SRTM3 GDEM only was used to derive data on the elevation of the lakes". However, following comment 1.3, we used ASTER GDEM in addition to SRTM to calculate slopes (Response to comment 1.3) and assess uncertainty. The text has been adjusted accordingly.

1.5. Comment: "The title is quite long. Try to shorten?"

Done: Assessment of Evolution and Risks of Glacier Lake Outbursts in the Djungarskiy Alatau, Central Asia using Landsat Imagery and Glacier Bed Topography Modelling.

Comment: L18, L132: contact with what?

Clarified: ice contact lakes.

Comment: L215: are their supraglacial lakes in the region? Is none of the ice contact-lakes ice dammed? Sure?

We confirm that there are no supraglacial and ice-dammed lakes in the region (although moraines supporting glaciers do contain ice). This is because glaciers in the study area are relatively small.

Comment: L299: sudden and complete drainage?

Changed to 'sudden and complete drainage'.

**Reviewer 2**

2.1.  Line 16: "In 2002 and 2014, 599 lakes with a combined area of $16.26\pm0.85$ km$^2$ and 636 lakes with a combined area of $17.35\pm0.92$ km$^2$ respectively were identified. The number of lakes and their combined area increased by 6.2 % and 6.6 % representing growth rates of 0.51 % a$^{-1}$ and 0.55 % a$^{-1}$."  Reviewers 1 and 2 suggested that changes in area are within the uncertainty of measurements.

We agree with this comment and the text has been corrected to reflect it: "Between 2002 and 2014, the number of lakes increased by 6.2% from 599 to 636 with a growth rate of 0.51 % a$^{-1}$. The combined areas were $16.26\pm0.85$ km$^2$ to $17.35\pm0.92$ km$^2$ respectively and the overall change was within the uncertainty of measurements".

2.2. Reviewer 2 suggested that a newly published paper by Petrov et al. (2017) on glacier lake inventory in Uzbekistan should be referred to in the Introduction and incorporated in the Discussion.

We included the paper by Petrov et al. (2017) to the review of the existing studies, Methods and Discussion.

2.3. Comment: "Lines 171-172. Channels 7, 4, 2 and panchromatic channel 8 of Landsat 7 are not the same as of Landsat 8. Their numbers are different and, in some cases, their wavelength bands also".

We have made a clarification in the text: "Therefore, lakes were mapped manually using channels 7, 4, 2 of Landsat 7 (Li and Sheng, 2012) and channels 3, 5, 7 of Landsat 8 as closest

to those of Landsat 7 (Table 1). The use of the panchromatic channel 8 with 15 m resolution, which requires manual mapping, enabled us to lower the threshold of digitisation from 2000 $m^2$ to 675 $m^2$."

We have added wavelengths of the Landsat 7 and Landsat 8 channels used in this study to Table 1.

2.4. Comment: "Lines 173-174. The threshold of digitization should 675 $m^2$ but not 700 $m^2$ (taking three 15×15 m pixels as lowest limit of lake identification)."

Correction has been made.

2.5. Comment: Line 346. The absolute vertical accuracy of ASTER GDEM2 of 17 m is given from (Meyer et al., 2011). But what is the vertical accuracy of ASTER GDEM2 relative to ice-free areas on topographic maps of Djungarskiy Alatau?

This is a valid comment. Only 1:50,000 scale maps were available to us (publishing information based on the higher-resolution maps is still problematic in Kazakhstan). Their relatively coarse resolution is unlikely to characterise the uncertainty of DEM in the mountains in a meaningful way. Instead, we compared elevations derived from both ASRTER and DEM to those of eight ground control points obtained using differential GPS and calculated RMSE to characterise uncertainty. The RMSE values were ±15 m and ±10 m for ASTER and SRTM respectively. This is very close to the values reported in literature. Only a small number of GCP was used because their collection commenced recently in this remote area. The RMSE values are given in the revised copy of the paper.

2.6. Comment: Line 666. I suggest to specify in the Conclusions: in which areas (basins) of the Djungarskiy Alatau the formation of a large number of new lakes is expected.

We have added the following to the Conclusions: "The highest number in the Aksu, Bien and Kora basins in the north-west and Lepsy basin in the north-east of the region."

All technical and stylistic corrections have been implemented.

---

## Author Response (AR1)

**Response to the comments made by Reviewers 1 and 2**

We have made all the corrections and implemented all suggestion made by the Reviewers. The detailed answers are given below. All changes are highlighted in the revised version of the manuscript.

**Reviewer 1**

1.1. Comment: Line 17 and at several places the 6.2% and 6.6% lake growth was confusing to me, because its significance level is unclear. Better focus (as done later, L365) on the number of lakes that showed significant growth, or similar. Note, changes without error bars, or some other indication of uncertainty are worthless. And I don't understand the mean value of uncertainty L200 and its meaning in this context.

Line 17: Reference to 6.6% increase was removed: "The combined areas were $16.26\pm0.85$ km$^2$ to $17.35\pm0.92$ km$^2$ respectively and the overall change was within the uncertainty of measurements."

We agree that the mean value of uncertainty in Line 200 is misleading. This phrase has been removed. Uncertainty values have been added throughout the text.

1.2. Comment: L273: threshold for what?

Line 279: This has been clarified: "Slopes steeper than 45º are considered as particularly dangerous in this regard (Alean, 1985; Bolch et al., 2011; Cook et al., 2016)."

1.3. L275: how good is the ASTER GDEM in particular over steep terrain, and how reliable is thus this slope threshold computation?

Firstly, see response to comment 1.4: SRTM3 GDEM was used in the original version.

Secondly, we have a limited number of ground-based geodetic measurements in the study region to quantify the uncertainty of DEM derived from satellite data. We have added RMSE values for both ASTER and SRTM based on several ground control points (GCP); see answers to comment 2.5. However, only a small number of GCP were available and we used a comparison between the two DEM to characterise uncertainty. The following text has been added:

Sections 3.1: "The void-filled SRTM3 GDEM (https://lta.cr.usgs.gov/SRTM1Arc) and ASTER GDEM2 (https://asterweb.jpl.nasa.gov/gdem.asp) with 30 m resolution were used to derive data on slope angles. SRTM3 DEM was used to derive elevations of the lakes. A reliable GDEM of the study area is essential for the assessment of thresholds for mass movements in

the vicinity of lakes and potential debris flow pathways. Both GDEM were shown to be suitable for assessments of slope angles and elevations in the Zailiiyskiy Alatau with limitations regarding smaller features (e.g. lateral moraines, deep gorges) and steep slopes (Bolch et al., 2011). To assess accuracy of both GDEM in the study region, elevations of eight ground control points obtained using differential GPS (DGPS) in the ice-free areas were compared with elevations derived from the GDEM. The RMSE values were ±10 m and ±15 m for SRTM3 and ASTER2 respectively. However, the number of ground-based measurements is insufficient to quantify uncertainty of slope estimation. Therefore, a comparison between slope angles derived from ASTER and SRTM was used to characterise uncertainty."

Section 3.4: "Slope values derived from ASTER GDEM exceeded those derived from SRTM: within the 500 m distance from Type 1 and 2 lakes, mean slope values and standard deviations were $16.9 \pm 4.8^{o}$ and $21.3 \pm 4.0^{o}$ for SRTM and ASTER respectively."

Section 5.5: "Results of modelling of outburst paths and mass movements depend on the quality of GDEM. The average difference between SRTM and ASTER GDEM was approximately $4^{o}$ for the slopes surrounding lakes with steeper slopes derived from ASTER GDEM. A detailed comparison of two GDEM and their validation will require extensive ground truth data which is currently unavailable. Due to the stringent criteria applied to the selection of dangerous lakes (i.e. distance to the infrastructure and average incline along the potential flood path line), the DEM uncertainty does not affect the 'first-order' identification of dangerous lakes presented here. However, a more accurate DEM and evaluation of its quality may be required for modelling of debris flow propagation."

.

1.4. Comment: L323: can you explain why you use the ASTER GDEM for slope assessments and SRTM for glacier bed estimation?

This is an unfortunate mistake which propagated through the text: SRTM3 GDEM was used for slope assessment and to calculate elevation bands in which lakes are positioned. The text has been corrected in Section 3.1 and throughout the text: "SRTM GDEM (https://lta.cr.usgs.gov/SRTM1Arc) and ASTER GDEM2 (https://asterweb.jpl.nasa.gov/gdem.asp) with 30 m resolution were used to derive data on slope angles while SRTM3 GDEM only was used to derive data on the elevation of the lakes". However, following comment 1.3, we used ASTER GDEM in addition to SRTM to calculate slopes (Response to comment 1.3) and assess uncertainty. The text has been adjusted accordingly.

1.5. Comment: "The title is quite long. Try to shorten?"

Done: Assessment of Evolution and Risks of Glacier Lake Outbursts in the Djungarskiy Alatau, Central Asia using Landsat Imagery and Glacier Bed Topography Modelling.

Comment: L18, L132: contact with what?

Clarified Line 133 (removed Line 18): ice contact lakes.

Comment: L215: are their supraglacial lakes in the region? Is none of the ice contact-lakes ice dammed? Sure?

We confirm that there are no supraglacial and ice-dammed lakes in the region (although moraines supporting glaciers do contain ice). This is because glaciers in the study area are relatively small.

Comment: L299: sudden and complete drainage?

Line 311: Changed to 'sudden and complete drainage'.

**Reviewer 2**

2.1. Line 16: "In 2002 and 2014, 599 lakes with a combined area of $16.26\pm0.85$ km$^2$ and 636 lakes with a combined area of $17.35\pm0.92$ km$^2$ respectively were identified. The number of lakes and their combined area increased by 6.2 % and 6.6 % representing growth rates of 0.51 % a$^{-1}$ and 0.55 % a$^{-1}$." Reviewers 1 and 2 suggested that changes in area are within the uncertainty of measurements.

Abstract: We agree with this comment and the text has been corrected to reflect it: "Between 2002 and 2014, the number of lakes increased by 6.2% from 599 to 636 with a growth rate of 0.51 % a$^{-1}$. The combined areas were $16.26\pm0.85$ km$^2$ to $17.35\pm0.92$ km$^2$ respectively and the overall change was within the uncertainty of measurements".

2.2. Reviewer 2 suggested that a newly published paper by Petrov et al. (2017) on glacier lake inventory in Uzbekistan should be referred to in the Introduction and incorporated in the Discussion.

We included the paper by Petrov et al. (2017) to the review of the existing studies (Lines 60-61), Methods (Line 239) and Discussion (Line 606).

2.3. Comment: "Lines 171-172. Channels 7, 4, 2 and panchromatic channel 8 of Landsat 7 are not the same as of Landsat 8. Their numbers are different and, in some cases, their wavelength bands also".

We have made a clarification in the text (Lines 183-186): "Therefore, lakes were mapped manually using channels 7, 4, 2 of Landsat 7 (Li and Sheng, 2012) and channels 3, 5, 7 of Landsat 8 as closest to those of Landsat 7 (Table 1). The use of the panchromatic channel 8 with 15 m resolution, which requires manual mapping, enabled us to lower the threshold of digitisation from 2000 $m^2$ to 675 $m^2$."

We have added wavelengths of the Landsat 7 and Landsat 8 channels used in this study to Table 1.

2.4. Comment: "Lines 173-174. The threshold of digitization should 675 $m^2$ but not 700 $m^2$ (taking three 15×15 m pixels as lowest limit of lake identification)."

Correction has been made (Line 185).

2.5. Comment: Line 346. The absolute vertical accuracy of ASTER GDEM2 of 17 m is given from (Meyer et al., 2011). But what is the vertical accuracy of ASTER GDEM2 relative to ice-free areas on topographic maps of Djungarskiy Alatau?

Assessment of vertical accuracy has been added using ground control points; Section 3.1 (See response to comment 1.3)

2.6. Comment: Line 666. I suggest to specify in the Conclusions: in which areas (basins) of the Djungarskiy Alatau the formation of a large number of new lakes is expected.

We have added the following to the Conclusions (Line 683): "The highest number in the Aksu, Bien and Kora basins in the north-west and Lepsy basin in the north-east of the region."

All technical and stylistic corrections have been implemented.